# Conformal Online Learning of Deep Koopman Linear Embeddings

**Ben Gao**
Université Jean Monnet Saint-Etienne, CNRS,
Institut d'Optique Graduate School, Inria,
Laboratoire Hubert Curien UMR 5516,
F-42023, SAINT- ETIENNE, France
ben.gao@univ-st-etienne.fr

**Jordan Patracone**[*]
Université Jean Monnet Saint-Etienne, CNRS,
Institut d'Optique Graduate School, Inria,
Laboratoire Hubert Curien UMR 5516,
F-42023, SAINT- ETIENNE, France
jordan.patracone@univ-st-etienne.fr

**Stéphane Chrétien**
Université Lyon 2
Laboratoire ERIC
Bron, France
stephane.chretien@univ-lyon2.fr

**Olivier Alata**
Université Jean Monnet Saint-Etienne, CNRS,
Institut d'Optique Graduate School,
Laboratoire Hubert Curien UMR 5516,
F-42023, SAINT- ETIENNE, France
olivier.alata@univ-st-etienne.fr

## Abstract

We introduce Conformal Online Learning of Koopman embeddings (COLoKe), a novel framework for adaptively updating Koopman-invariant representations of nonlinear dynamical systems from streaming data. Our modeling approach combines deep feature learning with multistep prediction consistency in the lifted space, where the dynamics evolve linearly. To prevent overfitting, COLoKe employs a conformal-style mechanism that shifts the focus from evaluating the conformity of new states to assessing the consistency of the current Koopman model. Updates are triggered only when the current model's prediction error exceeds a dynamically calibrated threshold, allowing selective refinement of the Koopman operator and embedding. Empirical results on benchmark dynamical systems demonstrate the effectiveness of COLoKe in maintaining long-term predictive accuracy while significantly reducing unnecessary updates and avoiding overfitting.

## 1 Introduction

Understanding the evolution of complex systems over time is essential in numerous disciplines, including robotics [Bruder et al., 2021], finance [Mann and Kutz, 2016], physics [Kaptanoglu et al., 2020], chemistry [Klus et al., 2020] and neuroscience [Chrétien et al., 2023]. A particularly powerful framework for this analysis is provided by operator-theoretic approaches, which recast nonlinear dynamics into linear evolution in function space. Among these, the *Koopman operator* [Budišić et al., 2012, Mauroy et al., 2020] plays a central role: it describes the progression of measurement functions (or observables) defined over the state space, yielding an infinite-dimensional linear representation of nonlinear systems. Its spectral decomposition offers a principled way to characterize the long-term behavior and underlying structure of the dynamics [Brunton et al., 2022].
Although the Koopman operator is infinite-dimensional, several numerical methods have been developed to approximate it in a finite-dimensional setting. Notable among these are Dynamic Mode Decomposition (DMD) [Rowley et al., 2009] and its nonlinear extensions, such as Extended DMD (EDMD) [Williams et al., 2015] and its variants (see [Jin et al., 2024] and references therein). Recent

---

[*] Research conducted within the context of the Inria–IIT Associate Team.

Table 1: Positioning of COLoKe with respect to the state-the-art.

| | No need for history | Online update | Adaptative embedding | Built-in reconstruction |
|---|---|---|---|---|
| ODMD [Zhang et al., 2019] | ✓ | ✓ | ✗ | ✓ |
| R-EDMD [Sinha et al., 2019, 2023] | ✗ | ✓ | ✗ | ✗ |
| DKLT [Hao et al., 2024] | ✓ | batch only | ✓ | ✗ |
| BatchOnline[Mazouchi et al., 2023] | ✓ | batch only | ✓ | ✗ |
| R-SSID [Loya and Tallapragada, 2024] | ✗ | batch only | ≈ | ✗ |
| OnlineAE [Liang et al., 2022] | ✓ | ✓ | ✓ | ✗ |
| COLoKe (ours) | ✓ | ✓ | ✓ | ✓ |

work has also proposed truncated Signature features rooted in path theory [Chrétien et al., 2025]. Another line of research consists of kernel learning formulations in reproducing kernel Hilbert spaces [Kostic et al., 2022, Hou et al., 2023]. In addition, deep learning techniques have been leveraged to learn expressive Koopman-invariant representations, using neural networks and autoencoder architectures to estimate significant observables [Li et al., 2017, Wehmeyer and Noé, 2018, Lusch et al., 2018, Yeung et al., 2019, Otto and Rowley, 2019]. For example, Xu et al. [2025b] addresses this by explicitly minimizing the spectral residual. The stochastic Koopman operator extends the classical Koopman framework to random dynamical systems, capturing the evolution of observables under stochastic influences [Črnjarić-Žic et al., 2020, Kostic et al., 2023, Xu et al., 2025a].

Most existing methods, however, operate in the *offline setting*, assuming access to the entire dataset in advance. Yet, in many applications—such as online monitoring, adaptive control, or real-time forecasting—data arrive sequentially, and the system may evolve in a non-stationary fashion [Korda and Mezić, 2018]. Several online methods have been proposed for this task. For instance, Online DMD [Zhang et al., 2019] and Online EDMD [Sinha et al., 2019, 2023] incrementally update Koopman operator estimates as new data arrives. However, these methods typically rely on linear observables or fixed dictionaries, limiting their expressiveness. More recent methods using neural networks (e.g. [Liang et al., 2022, Hao et al., 2024]) lack principled learning strategies and often rely on retraining the model using a fixed number of steps, regardless of whether the update is necessary. These limitations highlight the need for online learning strategies that are not only memory-efficient but also adaptive in order to update models only when required by the incoming data.

We address this need by introducing *Conformal Online Learning of Koopman embeddings (COLoKe)*, whose high-level principle is sketched in Figure 1. Our approach combines deep Koopman representation learning with a novel repurposing of conformal prediction principles to decide when to adapt the model as data arrive in a streaming fashion. Updates are triggered only when necessary, reducing both computational burden and overfitting. This contributes to the growing body of work on online Koopman learning [Sinha et al., 2023, Loya and Tallapragada, 2024, Mazouchi et al., 2023]. Our method differs by enabling adaptive embeddings, built-in reconstruction, and real-time updates while remaining memory-efficient (see Table 1). To the best of our knowledge, COLoKe is the first principled approach for online learning driven by conformal-based updates.

**Contributions.** In summary, our contributions are: (i) We propose a novel online learning framework that leverages conformal prediction to guide adaptive model updates; (ii) We instantiate this framework in the context of Koopman operator learning, enabling expressive online regression of nonlinear dynamical systems through deep data-adaptive embeddings; (iii) We provide a theoretical analysis establishing a dynamic regret bound under mild assumptions.

**Outline.** The rest of the paper is organized as follows. In Section 2, we present the mathematical background on Koopman operator theory and conformal prediction, which form the foundation of our approach. Section 3 introduces our main method and details how a conformal-based update strategy permits to update model parameters adaptively in an online fashion. Section 4 provides empirical validation on a range of benchmark dynamical systems, comparing COLoKe to existing online Koopman learning methods. Proofs and implementation details are deferred to the appendix.

## 2 Mathematical background

In this section, we present the two key components of our method, namely the Koopman operator and the conformal prediction framework.

## 2.1 Data-driven learning of the Koopman operator

Let a measurable space $(\mathcal{X}, \Sigma_{\mathcal{X}})$, with $\mathcal{X} \subset \mathbb{R}^d$ and $\Sigma_{\mathcal{X}}$ a Borel $\sigma$-algebra on $\mathcal{X}$, and let $T \colon \mathcal{X} \to \mathcal{X}$ be a measurable, time-invariant, deterministic map. Hereafter, we consider a discrete-time autonomous dynamical system governed by the iteration rule $x_{t+1} = T(x_t)$ for all $t \in \mathbb{N}$, which describes the evolution of the system as a sequence of states $\{x_t\}_{t \in \mathbb{N}}$ entirely determined by the initial condition $x_0 \in \mathcal{X}$ and the update rule $T$. A classical approach to analyze such kind of nonlinear dynamical systems is through the *Koopman operator* formalism. Rather than studying the trajectories in state space directly, the Koopman approach lifts the dynamics to an infinite-dimensional space of observables $\mathcal{F}$ (e.g., $L^2(\mathcal{X}, \mu)$ for some Borel measure $\mu$). The Koopman operator $\mathcal{K} \colon \mathcal{F} \to \mathcal{F}$ is defined by

$$(\mathcal{K}f)(x) = f(T(x)), \quad \forall f \in \mathcal{F}, \forall x \in \mathcal{X}, \tag{1}$$

which describes the evolution of observables along trajectories of the system. Importantly, $\mathcal{K}$ is a linear operator, even when $T$ is nonlinear, making it a powerful tool for the spectral analysis of nonlinear dynamics [Brunton et al., 2022]. In particular, if $\varphi \in \mathcal{F}$ is an eigenfunction of $\mathcal{K}$ with eigenvalue $\lambda \in \mathbb{C}$, i.e., $\mathcal{K}\varphi = \lambda\varphi$, then along a trajectory $(x_t)$, the observable evolves linearly: $\varphi(x_t) = \lambda^t \varphi(x_0)$. It follows that, when a set of eigenfunctions $\{\varphi_1, \ldots, \varphi_L\}$ defines an injective embedding of the state space, the system can be linearized via the coordinate transformation $x \mapsto (\varphi_1(x), \ldots, \varphi_L(x))$. In this lifted space, the nonlinear dynamics evolve linearly, providing a compelling framework for Koopman-based analysis and control [Korda and Mezić, 2018, Mauroy et al., 2020].

In many real-world scenarios, the transition map $T$ governing the dynamics is unknown or inaccessible, and we must instead rely on observed trajectories of the system $\{x_t\}_{t=0}^{N_t}$. This shift has led to the development of data-driven approximations of the Koopman operator $\mathcal{K}$. Motivated by the fact that Koopman eigenfunctions evolve linearly along trajectories, one typically seeks a set of observables $\{f_1, \ldots, f_m\} \subset \mathcal{F}$ that spans a subspace that is approximately invariant under the action of $\mathcal{K}$. In the ideal setting where $\mathcal{S} = \text{span}(f_1, \ldots, f_m)$ is invariant under $\mathcal{K}$, i.e., $\mathcal{K}f \in \mathcal{S}$ for all $f \in \mathcal{S}$, the restriction of $\mathcal{K}$ to $\mathcal{S}$ admits an exact representation by a finite-dimensional matrix $K \in \mathbb{C}^{m \times m}$, and the evolution of observables in this subspace is governed by the linear relation

$$\Phi(x_{t+1}) = K\Phi(x_t), \tag{2}$$

where $\Phi(x) = [f_1(x), \ldots, f_m(x)]^\top$ denotes the lifted representation of the state. This insight motivates the search for low-dimensional Koopman-invariant subspaces that admit such linear representations. Methods like EDMD [Williams et al., 2015] approximate this setting by fixing a dictionary $\{f_i\}_{i=1}^m$, but their performance is limited by the expressiveness and suitability of the chosen observables. To overcome this limitation, recent approaches [Takeishi et al., 2017, Lusch et al., 2018, Yeung et al., 2019, Otto and Rowley, 2019] propose to learn both the feature map $\Phi$ and the linear operator $K$ jointly using neural networks, leading to *deep Koopman embeddings*.

## 2.2 Conformal prediction for online data

Conformal prediction [Vovk et al., 1999, 2005, Romano et al., 2019, Angelopoulos and Bates, 2021] is a distribution-free framework for uncertainty quantification that constructs valid prediction sets with finite-sample guarantees.

Given past input–output pairs $\{(x_i, y_i)\}_{i=1}^{t-1}$, a model produces a prediction $\hat{y}_t$ for a new input $x_t$, and assigns a conformity score $s(x_t, y)$ to each candidate output $y$. The conformal prediction set is defined as $C_t = \{y \in \mathcal{Y} \mid s(x_t, y) \leq q_t\}$, where $q_t$ is a quantile calibrated to ensure $\mathbb{P}(y_t \in C_t) \geq 1 - \alpha$. The set $C_t$ is thus interpreted as a set of plausible outputs: it contains all candidate values of $y$ that are deemed sufficiently "conformal" (i.e., not too surprising) with respect to the current model and past observations. While this framework is powerful and requires no distributional assumptions beyond exchangeability, it is not directly applicable to time series or online learning, where data are typically non-exchangeable. In such settings, standard conformal methods may yield miscalibrated or overly conservative intervals. Addressing this challenge has motivated the development of adaptive conformal approaches that can track distribution shifts over time [Gibbs and Candes, 2021, Xu et al., 2021, Angelopoulos et al., 2023]. In particular, Conformal PID Control was recently introduced in Angelopoulos et al. [2023] as a dynamically calibrated version of conformal prediction. Here, "PID" refers to the use of *Proportional*, *Integral*, and (optionally) *Derivative* feedback terms—standard components in control theory—used to adaptively adjust the prediction threshold. Rather than fixing the quantile threshold $q_t$ in advance, the method updates it online in response to conformity violations.

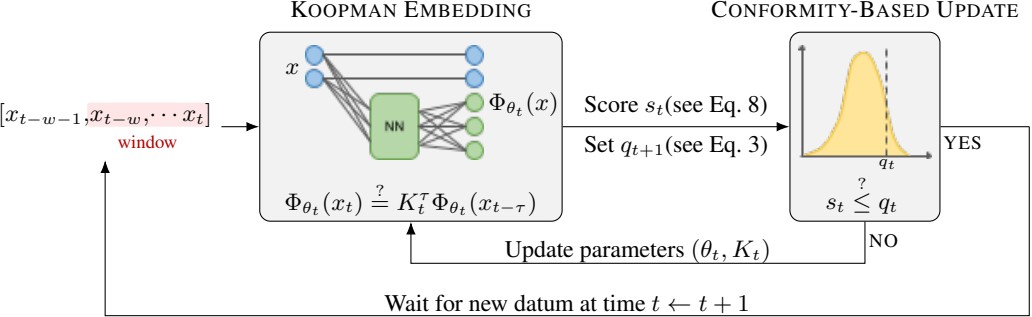

Figure 1: Schematic representation of COLoKe. The model receives a rolling window of observations, lifts them via a partially-learned feature map, computes a conformity score based on multi-step prediction error, and updates its parameters only if the score exceeds a conformal threshold.

Among its variants, we consider the conformal PI control scheme: after observing whether $y_t \in C_t$, a binary error signal $e_t = \mathbf{1}\{y_t \notin C_t\}$ is computed, and the threshold is adjusted via:

$$q_{t+1} = q_t + \underbrace{\gamma(e_t - \alpha)}_{\text{Proportional term (P)}} + \underbrace{r_t\left(\sum_{i=1}^{t}(e_i - \alpha)\right)}_{\text{Integral term (I)}}, \quad \forall t \in \mathbb{N}, \tag{3}$$

where $\gamma > 0$ is a learning rate and $r_t$ is a nonlinear saturation function [Angelopoulos et al., 2023] acting as an integral correction term. This PI-style control loop allows the prediction region to adaptively expand or contract to maintain the desired coverage. Under mild regularity conditions, it ensures that the empirical coverage converges, i.e., $\frac{1}{T}\sum_{t=1}^{T} e_t \to \alpha$ as $T \to \infty$.

In this paper, we revisit the conformal mechanism from a novel perspective. Rather than using conformal PI control to build uncertainty sets for $y_t$, we reinterpret the conformity score as a diagnostic tool for the model itself, in particular for deciding whether a Koopman embedding remains consistent over time or if it needs to be updated. This reinterpretation will be developed in Section 3.2.

## 3 Online conformal learning of deep Koopman embedding

### 3.1 From online learning...

We consider an online learning setting with bounded memory, where the goal is to incrementally learn a Koopman-invariant subspace from sequentially observed dynamics. Let $t \in \mathbb{N}$ denote the current time, and let $w \in \mathbb{N}^*$ be a fixed window size. At each time step $t$, we assume access to a finite buffer of recent observations $\mathcal{D}_t = \{x_{t-w}, \ldots, x_t\}$, which serves to incrementally update the current Koopman approximation.

At each time step $t$, we denote by $\Phi_{\theta_t} : \mathcal{X} \to \mathbb{C}^m$ the current feature map and by $K_t \in \mathbb{C}^{m \times m}$ the corresponding finite-dimensional Koopman operator. Following [Li et al., 2017], we also advocate to enforce interpretability and preserve part of the original state, by designing the feature map to include both the identity and a learnable nonlinear component, i.e.,

$$\Phi_{\theta_t}(x) = \left[x, \tilde{\Phi}_{\theta_t}(x)\right]^\top, \quad \forall x \in \mathcal{X} \tag{4}$$

where $\tilde{\Phi}_{\theta_t} : \mathcal{X} \to \mathbb{C}^{m-d}$ is a neural network with parameters $\theta_t$. This structure ensures that the lifted representation retains access to the state $x$ while learning an additional embedding $\tilde{\Phi}_{\theta_t}(x)$ from data. In order to capture temporal consistency within the buffer, we seek to minimize a multi-step prediction error over recent observations on $\mathcal{D}_t$. This leads to the following loss function used for online updates

**Definition 3.1** (Online Koopman training loss). At each time step, the parameters $(\theta_t, K_t)$ are updated by minimizing the multi-step prediction loss, i.e.,

$$\underset{(\theta_t, K_t)}{\text{minimize}} \left[ \mathcal{L}_t(\theta_t, K_t) := \sum_{(s,\tau) \in \mathcal{I}_t} \ell_{s,\tau}(\theta_t, K_t) \right], \tag{5}$$

where the index set $\mathcal{I}_t = \{(s, \tau) \in \mathbb{N}^2 \mid t - w \leq s < s + \tau \leq t\}$ collects all valid multi-step prediction pairs within the buffer $\mathcal{D}_t$, and each loss term is defined as

$$\ell_{s,\tau}(\theta_t, K_t) := \sum_{j=1}^{\tau} \left\| \Phi_{\theta_t}(x_{s+\tau}) - K_t^j \Phi_{\theta_t}(x_{s+\tau-j}) \right\|^2 \tag{6}$$

which accumulates the discrepancies between the lifted state at time $s + \tau$ and all intermediate predictions obtained by successively applying $K_t$ to earlier lifted states at times $\{s, \ldots, s + \tau - 1\}$.

This formulation is motivated by two design principles. First, as shown by Otto and Rowley [2019], multi-step prediction promotes the identification of persistent spectral modes and approximate Koopman eigenfunctions, thereby improving long-term prediction. Second, by explicitly including the state $x$ in the lifted representation (4), the model embeds a reconstruction constraint directly into the consistency loss. This coupling eliminates the need for a decoder, as prediction errors in the lifted space naturally reflect discrepancies in the original coordinates.

*Remark* 3.1 (Prediction conformity score). In particular, we have that

$$\ell_{t-w,w}(\theta_t, K_t) = \sum_{\tau=1}^{w} \left\| \Phi_{\theta_t}(x_t) - K_t^\tau \Phi_{\theta_t}(x_{t-\tau}) \right\|^2 \tag{7}$$

which quantifies the discrepancy between the *current* lifted state $\Phi_{\theta_t}(x_t)$ and its multi-step predictions from all previous states in the buffer. This term corresponds to predicting $x_t$ from each of the past $w$ observations using the learned Koopman operator $K_t$, and thus provides a direct measure of temporal consistency toward the *present*.

In principle, one could perform multiple optimization steps to minimize $\mathcal{L}_t(\theta_t, K_t)$ at each time $t$, thereby reducing the residual prediction error as much as possible within the local buffer. However, such an approach may lead to overfitting to recent data and degrade generalization. This behavior is typical of baseline online learning schemes that rely on fixed optimization schedules without accounting for model confidence or adaptation needs. To mitigate this issue, we propose a data-driven stopping rule inspired by conformal PID control, introduced in the next section, which adaptively decides whether additional updates are beneficial.

## 3.2 ... To conformal online learning

Assuming that the Koopman embedding, parametrized by $(\theta_{t-1}, K_{t-1})$, has been adequately trained on the previous buffer $\mathcal{D}_{t-1} = \{x_{t-1-w}, \ldots, x_{t-1}\}$, we now aim to determine whether it remains consistent with the newly observed state $x_t$. Rather than retraining unconditionally at each step, we initialize $(\theta_t, K_t) := (\theta_{t-1}, K_{t-1})$ and update the model only when the incoming data indicates a significant deviation from previously learned dynamics.

To assess this deviation, we rely on the *prediction conformity score* introduced in Remark 3.1, which measures the discrepancy between the current lifted state $\Phi_{\theta_t}(x_t)$ and its multi-step predictions from the past window. This score acts as a proxy for temporal alignment and forms the basis of our adaptive update rule. To formalize this idea, we define a score function $s_t$ by treating the prediction conformity score as a function of the test point $x \in \mathcal{X}$, with model parameters $(\theta_t, K_t)$ fixed:

$$s_t(x, (\theta_t, K_t)) := \sum_{\tau=1}^{w} \left\| \Phi_{\theta_t}(x) - K_t^\tau \Phi_{\theta_t}(x_{t-\tau}) \right\|^2. \tag{8}$$

Note the difference with (7), in the sense that $s(x, (\theta_t, K_t))$ is a function of the additional variable $x \in \mathcal{X}$. The score $s(x, (\theta_t, K_t))$ is instrumental in defining the prediction set as introduced now. Inspired by the conformal prediction interval (PI) framework described in Section 2.2, we define a prediction set as

$$C_t = \{x \in \mathcal{X} \mid s_t(x, (\theta_t, K_t)) \leq q_t\}, \tag{9}$$

**Algorithm 1** Conformal Online Learning of Koopman embeddings (COLoKe)

---

**Require:** Buffer size $w$, initial parameters $(\theta_{w-1}, K_{w-1})$, step-size $\eta > 0$
1: Initialize conformity threshold $q_w$
2: **for** $t = w, w+1, \ldots$ **do**
3:      Observe new state $x_t$ and update buffer $\mathcal{D}_t = \{x_{t-w}, \ldots, x_t\}$
4:      Set $(\theta_t, K_t) \leftarrow (\theta_{t-1}, K_{t-1})$
5:      Compute prediction conformity score $s_t \leftarrow \ell_{t-w,w}(\theta_t, K_t)$          ▷ See Eq. (7)
6:      Update threshold: $q_{t+1} \leftarrow \mathrm{ConformalPI}(q_t)$      ▷ See Eq. (3) with $e_t = \mathbf{1}\{s_t > q_t\}$
7:      **while** $s_t > q_t$ **do**
8:          Perform a gradient-based step: $(\theta_t, K_t) \leftarrow (\theta_t, K_t) - \eta \nabla_{\theta, K} \mathcal{L}_t(\theta_t, K_t)$    ▷ See Eq. (5)
9:          Recompute $s_t \leftarrow \ell_{t-w,w}(\theta_t, K_t)$
10:      **end while**
11: **end for**

---

where $q_t > 0$ is a calibration threshold that controls the conformity level. In the standard conformal prediction setting, the set $C_t$ serves as a prediction region for the next state $x_t$. This interpretation treats $C_t$ as a $(1 - \alpha)$-confidence region in which the next observation is expected to fall, based on past conformal scores.

In our setting, however, we use this prediction set in a novel way: not for uncertainty quantification, but as a decision rule for model adaptation. Specifically, if the newly observed state $x_t$ lies outside $C_t$, the prediction error is considered too large relative to past conformity, and an update of $(\theta_t, K_t)$ is triggered. Otherwise, the model is retained without further training. This repurposing of conformal principles provides a lightweight, data-driven mechanism for online learning.

Crucially, while classical conformal prediction evaluates the conformity of new *states*, our approach shifts perspective to evaluate the conformity of *parameter configurations*. That is, rather than asking whether a new observation aligns with a fixed model, we ask whether there exists any parameter pair $(\theta, K)$ under which the current *observed* state $x_t$ is conformal.

In this spirit, and since we are not primarily interested in constructing a prediction interval for $x_t$, we introduce the novel notion of prediction score set.

**Definition 3.2** (Prediction score set). Given a newly observed state $x_t$ and a conformity threshold $q_t > 0$, the *prediction score set* at time $t$ is defined as

$$S_t = s_t(x_t, \mathrm{Param}_t), \tag{10}$$

where $\mathrm{Param}_t = \{(\theta, K)$ such that $s = s_t(x_t, (\theta, K)) \equiv \ell_{t-w,w}(\theta, K) \leq q_t\}$. This set contains all prediction scores attainable at $x_t$ by Koopman models that satisfy the current calibration constraint.

In this view, if the current model satisfies $\ell_{t-w,w}(\theta_t, K_t) \in S_t$ (or equivalently, $\ell_{t-w,w}(\theta_t, K_t) \leq q_t$), it is deemed temporally consistent and it is retained. Conversely, if $\ell_{t-w,w}(\theta_t, K_t) \notin S_t$ (i.e., $\ell_{t-w,w}(\theta_t, K_t) > q_t$), the current model is no longer consistent with past dynamics, and an update of $(\theta_t, K_t)$ is triggered. Note that, by design, there is no need to construct $\mathrm{Param}_t$ explicitly. We refer to the resulting adaptive online learning scheme, driven by conformity-based update decisions, as *Conformal Online Learning of Koopman embeddings* (COLoKe). The full procedure implementing the principles of conformal-based online learning is summarized in Algorithm 1 and sketched in Fig. 1. We provide below a preliminary bound for the dynamic regret of COLoKe.

**Theorem 3.3** (Dynamic regret of COLoKe). *Let $(\theta_t, K_t)$ be the parameters produced by Algorithm 1 and let $(\theta_t^*, K_t^*) \in \mathrm{argmin}_{(\theta, K)} \mathcal{L}_t(\theta, K)$ denote any time-dependent optimal model minimizing the loss at step $t$. Further assume:*

*(A1) Each $\mathcal{L}_t$ is $L$-smooth with $\|\nabla \mathcal{L}_t(\theta, K)\| \leq B$;*

*(A2) The oracle path has bounded total variation and squared variation:*

$$V_T := \sum_{t=1}^{T} \|(\theta_{t+1}^*, K_{t+1}^*) - (\theta_t^*, K_t^*)\| < \infty, \quad S_T := \sum_{t=1}^{T} \|(\theta_{t+1}^*, K_{t+1}^*) - (\theta_t^*, K_t^*)\|^2 < \infty;$$

*(A3) The conformity thresholds satisfy $\sum_{t=1}^{T} q_t \leq \mathcal{O}(\alpha h(T))$ for some sublinear, nonnegative, nondecreasing function $h$;*

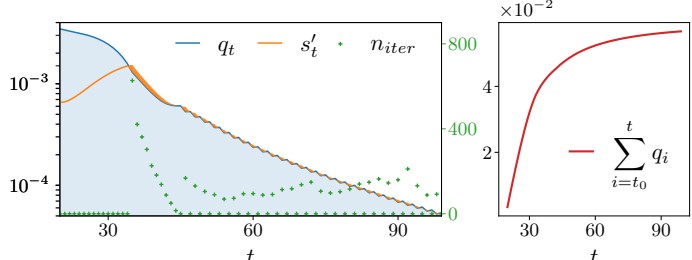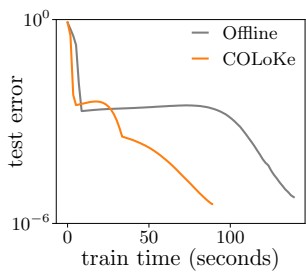

(a) Left: Adaptive update behavior with threshold $q_t$, score $s_t$, and update counts; Right: sublinear growth supporting (**A3**) in Theorem 3.3.

(b) COLoKe vs. offline Koopman learning

Figure 2: Illustration and empirical support for COLoKe's adaptive learning strategy.

*Then the dynamic regret satisfies:* $\sum_{t=1}^{T} \left[ \mathcal{L}_t(\theta_t, K_t) - \mathcal{L}_t(\theta_t^*, K_t^*) \right] \leq \mathcal{O}\left( \alpha h(T) + V_T + S_T \right).$

*Proof.* The proof is deferred to Appendix A. □

Assumptions (A1)–(A2) are standard in online learning and naturally satisfied in our setting. The loss $\mathcal{L}_t$ is smooth in both $\theta$ and $K$, provided that the neural network $\tilde{\Phi}_\theta$ employs smooth activations. The bounded variation of the dynamic oracle (A2) reflects slowly evolving or piecewise-stationary dynamics, which commonly arise in practice. The key nonstandard assumption is (A3), which bounds the cumulative conformity thresholds $q_t$, and which follows from our use of conformal PI. More specifically, the function $h$ comes from the saturation function $r_t$ in the update rule (3), as designed in Angelopoulos et al. [2023]. While we state (A3) as an assumption, it can be viewed as an empirical hypothesis: in regimes with stable distributions and smoothly adapting models, conformity thresholds decrease rapidly, leading to sublinear accumulation. This behavior is consistently observed across our experiments (see Fig. 2a), and deriving it remains an important direction for future work. For intuition, imagine a hypothetical scenario where (A3) does not hold, for instance where the cumulative sum grows linearly. This would imply that the scores stay roughly constant over time. From a conformal prediction standpoint, this means that predictive uncertainty does not shrink as more data arrives, indicating the model fails to learn a better representation. Such behavior may arise in non-autonomous systems under frequent abrupt changes, or when the model lacks the expressivity to learn the dynamics. While the former case is out of the scope of the present paper, the second case can be mitigated by augmenting the dimension of the lifted space.

Our proposed conformal online learning approach can be applied to more general online non-convex learning settings beyond Koopman framework. An important contribution to online non-convex learning is due to Suggala and Netrapalli [2020], who achieves optimal regret even under non-convex and adversarial losses by assuming access to an offline oracle. Subsequent work on dynamic regret explores variants to handle changing environments [Xu and Zhang, 2024]. Although these approaches specify how to update model parameters at each iteration, they do not consider *when* such updates should take place or *to what extent*. Our approach provides a statistically principled answer to these questions.

## 4 Numerical experiments

We now conduct experiments on multiple datasets derived either from solving differential equations associated with canonical dynamical systems, or from real-world sequential measurements. For the sake of reproducibility, all datasets and methods can be found at https://github.com/ben2022lo/COLoKe, and we report full implementation details as well as complementary numerical studies in the supplementary material.

### 4.1 Illustration and validation

To build intuition about our conformity-based update mechanism, we begin by illustrating how COLoKe behaves in a controlled setting. Then, we assess whether it (i) accurately recovers the

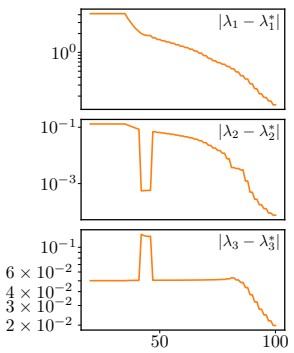

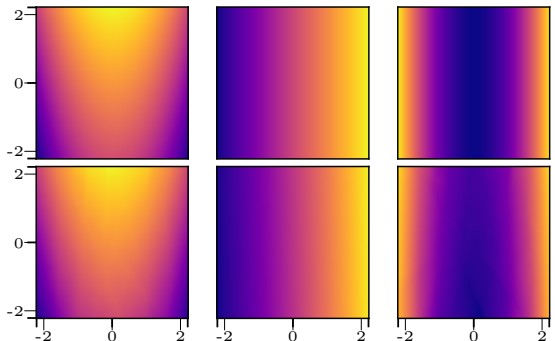

(a) Eigenvalues estimation error as functions of the dynamic $t$.

(b) Eigenfunctions $\{\varphi_1, \varphi_2, \varphi_3\}$ (from left to right) corresponding to the oracle (top) and estimated by COLoKe (bottom).

Figure 3: Convergence of the Koopman eigenvalue and eigenfunction estimates in the online setting.

underlying Koopman model, and (ii) achieves predictive performance comparable to offline learning approaches. Beyond overall accuracy, we are particularly interested in evaluating the quality of the estimated spectral properties. To this end, we consider the following analytically tractable system with known Koopman eigenvalues and eigenfunctions.

**Setting.** To validate the spectral accuracy of our online algorithm, we consider a benchmark system with analytically known Koopman eigenvalues and eigenfunctions [Brunton et al., 2016, 2022]:

$$\forall t \in \mathbb{R}_+, \quad \begin{cases} \dot{u}(t) & = au(t), \\ \dot{v}(t) & = b(v(t) - u^2(t)), \end{cases} \tag{11}$$

where $\dot{u}$ denotes the time derivative of $u$. For $a = -0.05$ and $b = -1$, the system admits a single attracting manifold $v = u^2$. The Koopman spectrum contains the eigenvalues $\{\lambda_1^* = -1, \lambda_2^* = -0.05, \lambda_3^* = -0.1\}$ with known analytical eigenfunctions. To generate data, we simulate trajectories of the state vector $x_t = [u(t), v(t)]^\top$, using a fixed integration step $0.01$. We produce 1000 training trajectories of length 100 by sampling initial conditions uniformly from $[-2, 2]^2$, and similarly generate 1000 additional trajectories for testing.

**Illustration and role of conformity-based updates.** Fig. 2a (left) depicts the evolution of the calibration threshold $q_t$ (blue line) and the associated prediction score set $S_t$ (shaded blue region). When the model yields a nonconformant score $s_t > q_t$ (top orange), updates are triggered until the score becomes conformant $s_t \leq q_t$ (bottom orange). The number of updates (green crosses) varies across time, reflecting when and how much the model must adjust to maintain consistency. As training progresses, the steady decay of $q_t$ reflects growing confidence and temporal alignment of the model with the data. This increased accuracy tightens the score set $S_t$, making conformity harder to achieve—yet this is a desirable outcome, as it ensures high-precision adaptation. Importantly, the number of updates remains controlled, showing that conformity can be maintained without overfitting or instability. To complement these observations, Fig. 2a (right) reports the cumulative thresholds, which exhibit sublinear growth of order $\mathcal{O}(\sqrt{T})$, thereby supporting Assumption (**A3**) in Theorem 3.3. Further analysis about how the conformal update-trigger mechanism adapts to different dynamical systems can be found in Appendix C.

**Spectral properties.** We monitor the spectral behavior of the learned Koopman operator by diagonalizing $K_t$ at each time step and tracking its eigenvalues. Although eigenvalues may be complex in general, the true values in this setting are real. Fig. 3a reports the estimation errors over time, showing that COLoKe recovers the correct spectrum, the estimated eigenvalues stabilize around their true values. At the end of training, we obtain real-valued eigenvalues $\{-1.0091, -0.04996, -0.1001\}$, which closely match the ground-truth. The estimated eigenfunctions, shown in Fig. 3b, align with the oracle up to a scalar factor.

**Comparison with offline Koopman learning.** To assess the computational efficiency of COLoKe, we compare it with an offline deep Koopman model trained on full trajectories using the same neural architecture. Figure 2b reports the test error as a function of training time (in seconds), measured on an NVIDIA RTX 2000 ADA GPU. Note that, the test error is evaluated on a separate

Table 2: Numerical evaluation of synthetic and real (indicated by $\star$) datasets. For synthetic datasets, the first and second line corresponds to generalization error and online error respectively.

| | ODMD | OEDMD | OnlineAE | OLoKe | COLoKe |
|---|---|---|---|---|---|
| Single attractor | $1.1 \cdot 10^{-3}$ $(\pm 3.6 \cdot 10^{-5})$ $4.6 \cdot 10^{-5}$ $(\pm 7.3 \cdot 10^{-7})$ | $2.5 \cdot 10^{-2}$ $(\pm 2.8 \cdot 10^{-4})$ $1.5 \cdot 10^{-2}$ $(\pm 4.7 \cdot 10^{-4})$ | $1.0 \cdot 10^{-2}$ $(\pm 7.7 \cdot 10^{-4})$ $7.4 \cdot 10^{-5}$ $(\pm 2.8 \cdot 10^{-5})$ | $2.1 \cdot 10^{-6}$ $(\pm 6.6 \cdot 10^{-7})$ $7.5 \cdot 10^{-6}$ $(\pm 2.5 \cdot 10^{-6})$ | $\mathbf{2.4 \cdot 10^{-7}}$ $(\pm \mathbf{3.6 \cdot 10^{-8}})$ $\mathbf{7.6 \cdot 10^{-7}}$ $(\pm \mathbf{9.6 \cdot 10^{-8}})$ |
| Duffing oscillator | $2.5 \cdot 10^{-4}$ $(\pm 7.8 \cdot 10^{-6})$ $1.9 \cdot 10^{-4}$ $(\pm 1.5 \cdot 10^{-6})$ | $6.8 \cdot 10^{-3}$ $(\pm 4.5 \cdot 10^{-3})$ $3.8 \cdot 10^{-3}$ $(\pm 3.3 \cdot 10^{-4})$ | $8.7 \cdot 10^{-3}$ $(\pm 2.5 \cdot 10^{-3})$ $2.0 \cdot 10^{-3}$ $(\pm 6.2 \cdot 10^{-4})$ | $5.5 \cdot 10^{-5}$ $(\pm 1.0 \cdot 10^{-5})$ $2.3 \cdot 10^{-4}$ $(\pm 4.0 \cdot 10^{-5})$ | $\mathbf{3.1 \cdot 10^{-6}}$ $(\pm \mathbf{2.3 \cdot 10^{-7}})$ $\mathbf{7.3 \cdot 10^{-5}}$ $(\pm \mathbf{1.9 \cdot 10^{-5}})$ |
| VdP oscillator | $2.1 \cdot 10^{-3}$ $(\pm 3.6 \cdot 10^{-5})$ $1.1 \cdot 10^{-3}$ $(\pm 4.7 \cdot 10^{-6})$ | $2.1 \cdot 10^{-3}$ $(\pm 3.2 \cdot 10^{-5})$ $1.1 \cdot 10^{-3}$ $(\pm 7.8 \cdot 10^{-6})$ | $1.7 \cdot 10^{-2}$ $(\pm 3.0 \cdot 10^{-3})$ $3.8 \cdot 10^{-3}$ $(\pm 1.0 \cdot 10^{-3})$ | $6.6 \cdot 10^{-4}$ $(\pm 1.5 \cdot 10^{-4})$ $9.2 \cdot 10^{-4}$ $(\pm 3.0 \cdot 10^{-4})$ | $\mathbf{3.8 \cdot 10^{-4}}$ $(\pm \mathbf{1.2 \cdot 10^{-5}})$ $\mathbf{6.0 \cdot 10^{-4}}$ $(\pm \mathbf{1.4 \cdot 10^{-4}})$ |
| Lorenz system | $2.7 \cdot 10^{-1}$ $(\pm 1.3 \cdot 10^{-3})$ $1.0 \cdot 10^{-1}$ $(\pm 5.8 \cdot 10^{-4})$ | $5.5 \cdot 10^{-1}$ $(\pm 2.2 \cdot 10^{-2})$ $2.7 \cdot 10^{-1}$ $(\pm 3.1 \cdot 10^{-2})$ | $5.9 \cdot 10^{-1}$ $(\pm 8.4 \cdot 10^{-2})$ $3.8 \cdot 10^{-2}$ $(\pm 2.6 \cdot 10^{-3})$ | $7.6 \cdot 10^{-3}$ $(\pm 1.8 \cdot 10^{-4})$ $4.7 \cdot 10^{-3}$ $(\pm 3.0 \cdot 10^{-4})$ | $\mathbf{6.5 \cdot 10^{-3}}$ $(\pm \mathbf{1.0 \cdot 10^{-4}})$ $\mathbf{3.3 \cdot 10^{-3}}$ $(\pm \mathbf{1.1 \cdot 10^{-4}})$ |
| ETD $\star$ | $1.2 \cdot 10^{-1}$ $(\pm 2.9 \cdot 10^{-1})$ | $1.5 \cdot 10^{-1}$ $(\pm 2.6 \cdot 10^{-1})$ | $7.9 \cdot 10^{-2}$ $(\pm 7.3 \cdot 10^{-2})$ | $9.7 \cdot 10^{-2}$ $(\pm 8.5 \cdot 10^{-2})$ | $\mathbf{7.3 \cdot 10^{-2}}$ $(\pm \mathbf{6.3 \cdot 10^{-2}})$ |
| EEG $\star$ | $8.3 \cdot 10^{-3}$ $(\pm 1.13 \cdot 10^{-2})$ | $8.0 \cdot 10^{-3}$ $(\pm 1.11 \cdot 10^{-2})$ | $1.24 \cdot 10^{-2}$ $(\pm 1.81 \cdot 10^{-2})$ | $8.8 \cdot 10^{-3}$ $(\pm 9.7 \cdot 10^{-3})$ | $\mathbf{7.8 \cdot 10^{-3}}$ $(\pm \mathbf{8.5 \cdot 10^{-3}})$ |
| Turbulence $\star$ | $5.5 \cdot 10^{-4}$ $(\pm 7.5 \cdot 10^{-4})$ | $7.3 \cdot 10^{-4}$ $(\pm 8.9 \cdot 10^{-4})$ | $6.4 \cdot 10^{-4}$ $(\pm 9.0 \cdot 10^{-4})$ | $4.8 \cdot 10^{-4}$ $(\pm 6.3 \cdot 10^{-4})$ | $\mathbf{4.7 \cdot 10^{-4}}$ $(\pm \mathbf{5.9 \cdot 10^{-4}})$ |

set of full trajectories, making it a reliable measure of generalization rather than step-wise prediction accuracy. While the offline method requires optimizing over the entire dataset, COLoKe incrementally adapts its model and reaches lower test error in significantly less time. The gap widens as training progresses, highlighting the advantage of online updates in terms of both speed and generalization. This experiment confirms that COLoKe delivers competitive predictive performance while being substantially more frugal computationally.

## 4.2 Comparison with online baselines

We now evaluate the benefits of COLoKe against state-of-the-art online approaches on a suite of benchmark dynamical systems, commonly used in Koopman and machine learning studies, and spanning a wide range of complexity.

**Datasets.** We consider dynamical systems with a single attractor (*Single Attractor*) [Brunton et al., 2016], two stable spirals and a saddle point (*Duffing oscillator*) [Williams et al., 2015], a limit cycle (*Van der Pol oscillator*) [Sinha et al., 2019], and a chaotic regime with a strange attractor (*Lorenz system*) [Kostic et al., 2022]. We complement them with three real-world datasets: the Electricity Transformer Dataset (*ETD*) [Zhou et al., 2021], the EEG Motor Movement/Imagery Dataset [Schalk et al., 2004, Goldberger et al., 2000] and a real turbulence dataset *CASES-99* [Earth Observing Laboratory, 1999]. These datasets introduce additional challenges such as noise and potential distribution shifts. Full details are provided in Appendix B.1.

**Baselines.** We compare COLoKe against the online learning strategies listed in Table 1. While the table includes both online and batch-based methods, our evaluation focuses only on those compatible with fully streaming, one-sample-at-a-time updates, excluding methods that require access to data batches. For R-EDMD [Sinha et al., 2019, 2023], since the radial basis function dictionary requires the full data to estimate the centers (see Table 1), we use a polynomial dictionary. And the reconstruction matrix is estimated with the current buffer by regularized pseudo-inverse. This *purely online* variant

is coined Online EDMD (*OEDMD*). For completeness, we also include a variant of COLoKe, referred to as *OLoKe*, in which the conformal update rule is dropped and replaced by a standard strategy: at each time step, the parameters are updated using a fixed number of gradient steps upon receiving a new sample. Implementation details are reported in Appendix B.2.

**Metrics.** We report two complementary metrics to evaluate model performance. The *generalization error* measures one-step prediction error on unseen trajectories. The *online prediction error*, on the other hand, quantifies performance during streaming inference by averaging the training prediction loss over time as new data arrives. For real datasets, models are evaluated on a single trajectory and only online prediction error are computed. We also report the computation time of each method across all datasets in Appendix D.

**Results.** Table 2 reports the performance averaged over 5 random splits of 2000 trajectories for simulated datasets, along with standard deviations of the means. Several important observations emerge. First, OEDMD underperforms ODMD, primarily due to two factors: the reconstruction matrix is estimated online, which limits accuracy; the appropriate choice of dictionary (i.e., polynomial degree) is unknown and not adaptively selected. Second, COLoKe, combining a flexible neural architecture with a principled update scheme, consistently outperforms all baselines across both synthetic and real-world datasets. In particular, it systematically outperforms its fixed-step counterpart OLoKe, demonstrating the benefit of adaptive model refinement through conformity-based updates. On the chaotic Lorenz system, (C)OLoKe achieves an improvement of nearly two orders of magnitude over the baselines, highlighting its effectiveness in capturing highly complex and sensitive nonlinear dynamics. Third, on non-autonomous real-world datasets, COLoKe achieves the best online performance, highlighting the capacity of conformal PI control to dynamically adjust to distribution shift. Altogether, these results show that COLoKe combines expressive modeling with principled online adaptation, making it a strong and reliable choice for real-time learning in complex, ever-evolving dynamical environments.

**Efficiency of adaptive updates.** To further highlight the benefits of conformal-based updates, we compared COLoKe to OLoKe trained with fixed update budgets of 1, 5, 10, 50, 100, and 150 iterations on the high-dimensional real EEG dataset. Unlike these fixed-iteration baselines, COLoKe does not require manual tuning: its updates are triggered automatically based on conformity. As shown in Figure 4, it achieves both lower online error and substantially lower execution time than all fixed-budget variants of OLoKe. This demonstrates that adaptive conformal triggering not only prevents unnecessary computations, but also outperforms any fixed-step strategy without the need to search for an optimal iteration.

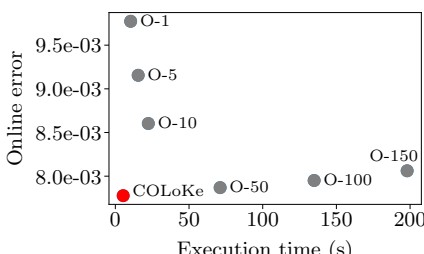

Figure 4: Pareto front comparing COLoKe to fixed-step OLoKe variants.

## 5 Conclusion

We have introduced COLoKe, a principled framework for online learning of Koopman-invariant representations, where conformal prediction is repurposed to adaptively trigger updates only when the model becomes temporally inconsistent. By moving beyond fixed-step updates, COLoKe achieves accurate and efficient learning of linear embeddings for nonlinear dynamics when data arrive sequentially. This work opens several promising directions for future research. On the theoretical front, our results highlight the need for a more comprehensive understanding of conformity-based online learning, with potential relevance well beyond Koopman operator estimation. The main limitation of our analysis is assumption (**A3**), whose validity remains open although being empirically supported; future work could aim to derive it from first principles for autonomous systems. On the methodological front, extending COLoKe to non-autonomous systems [Gao et al., 2025] could further broaden its applicability and help unlock practical Koopman learning in real-time, resource-constrained, and dynamically evolving environments.

**Acknowledgements** This work was sponsored by a public grant overseen by Auvergne-Rhône-Alpes region, Grenoble Alpes Metropole and BPIFrance, as part of project I-Démo Région "Green AI".

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

# A Proof of Theorem 3.3

For the reader's convenience, we restate Theorem 3.3 below.

**Theorem 3.3** *Let $(\theta_t, K_t)$ be the parameters produced by Algorithm 1 and let $(\theta_t^*, K_t^*) \in \text{argmin}_{(\theta, K)} \mathcal{L}_t(\theta, K)$ denote any time-dependent optimal model minimizing the loss at step $t$. Further assume:*

*(A1) Each $\mathcal{L}_t$ is L-smooth with $\|\nabla \mathcal{L}_t(\theta, K)\| \leq B$;*
*(A2) The oracle path has bounded total variation and squared variation:*

$$V_T := \sum_{t=1}^{T} \|(\theta_{t+1}^*, K_{t+1}^*) - (\theta_t^*, K_t^*)\| < \infty, \quad S_T := \sum_{t=1}^{T} \|(\theta_{t+1}^*, K_{t+1}^*) - (\theta_t^*, K_t^*)\|^2 < \infty;$$

*(A3) The conformity thresholds satisfy $\sum_{t=1}^{T} q_t \leq \mathcal{O}(\alpha h(T))$ for some sublinear, nonnegative, nondecreasing function $h$;*

*Then the dynamic regret satisfies: $\sum_{t=1}^{T} [\mathcal{L}_t(\theta_t, K_t) - \mathcal{L}_t(\theta_t^*, K_t^*)] \leq \mathcal{O}(\alpha h(T) + V_T + S_T)$.*

*Proof.* Since $\ell_{t-w,w}(\theta_t, K_t) \leq q_t$, we have

$$\mathcal{L}_t(\theta_t, K_t) = \sum_{(s,\tau) \in \mathcal{I}_t} \ell_{s,\tau}(\theta_t, K_t) \leq |\mathcal{I}_t| \cdot \ell_{t-w,w}(\theta_t, K_t) \leq \frac{w(w+1)}{2} \cdot q_t.$$

Let $c = \frac{w(w+1)}{2}$, then we have:

$$\mathcal{L}_t(\theta_t, K_t) - \mathcal{L}_t(\theta_t^*, K_t^*) \leq c \cdot q_t - \mathcal{L}_t(\theta_t^*, K_t^*).$$

Let us define the error:

$$\epsilon_t := c \cdot q_t - \mathcal{L}_t(\theta_t^*, K_t^*).$$

We decompose $\epsilon_t$ as:

$$\epsilon_t = \underbrace{c \cdot q_t - \mathcal{L}_t(\theta_{t+1}^*, K_{t+1}^*)}_{(a)} + \underbrace{\mathcal{L}_t(\theta_{t+1}^*, K_{t+1}^*) - \mathcal{L}_t(\theta_t^*, K_t^*)}_{(b)}.$$

**Step a:** By assumption (A3),

$$\sum_{t=1}^{T} (c \cdot q_t - \mathcal{L}_t(\theta_{t+1}^*, K_{t+1}^*)) \leq c \cdot \sum_{t=1}^{T} q_t \leq \mathcal{O}(\alpha h(T)).$$

**Step b:** We now bound the drift term due to the oracle movement.

Let $z_t^* := (\theta_t^*, K_t^*)$ and $z_{t+1}^* := (\theta_{t+1}^*, K_{t+1}^*)$. Since $\mathcal{L}_t$ is L-smooth with $\|\nabla \mathcal{L}_t\| \leq B$, we use the standard smoothness bound:

$$\mathcal{L}_t(z_{t+1}^*) - \mathcal{L}_t(z_t^*) \leq \langle \nabla \mathcal{L}_t(z_t^*), z_{t+1}^* - z_t^* \rangle + \frac{L}{2} \|z_{t+1}^* - z_t^*\|^2$$

$$\leq \|\nabla \mathcal{L}_t(z_t^*)\| \cdot \|z_{t+1}^* - z_t^*\| + \frac{L}{2} \|z_{t+1}^* - z_t^*\|^2.$$

Using $\|\nabla \mathcal{L}_t(z_t^*)\| \leq B$, define $\Delta_t := \|z_{t+1}^* - z_t^*\|$, so:

$$\mathcal{L}_t(z_{t+1}^*) - \mathcal{L}_t(z_t^*) \leq B\Delta_t + \frac{L}{2}\Delta_t^2.$$

Summing over $t$ from 1 to $T$:

$$\sum_{t=1}^{T} \mathcal{L}_t(z_{t+1}^*) - \mathcal{L}_t(z_t^*) \leq B\sum_{t=1}^{T}\Delta_t + \frac{L}{2}\sum_{t=1}^{T}\Delta_t^2 = BV_T + \frac{L}{2}S_T = \mathcal{O}(V_T + S_T).$$

Combining both contributions:

$$\sum_{t=1}^{T} [\mathcal{L}_t(\theta_t, K_t) - \mathcal{L}_t(\theta_t^*, K_t^*)] \leq \mathcal{O}(\alpha h(T)) + \mathcal{O}(V_T + S_T) = \mathcal{O}(\alpha h(T) + V_T + S_T).$$

$\square$

## B  Experimental settings

### B.1  Datasets and metrics

#### B.1.1  Simulated data

We evaluate performance on four synthetic datasets generated by solving ordinary differential equations (ODEs). For each dynamic, we simulate 2000 trajectories and construct 5 random train-test splits $\{(\mathcal{I}_k^{\text{train}}, \mathcal{I}_k^{\text{test}})\}_{k=1}^5$. For each split, models are trained using the training trajectories while computing the online prediction error

$$\varepsilon_k = \frac{1}{|\mathcal{I}_k^{\text{train}}|} \sum_{i \in \mathcal{I}_k^{\text{train}}} \frac{1}{T - t_0} \sum_{t=t_0+1}^{T} \|x_t^{(i)} - \text{Model}_{t-1}(x_{t-1}^{(i)})\|^2. \tag{12}$$

After the training, models are evaluated on the held-out trajectories to compute generalization error

$$\xi_k = \frac{1}{|\mathcal{I}_k^{\text{test}}|} \sum_{i \in \mathcal{I}_k^{\text{test}}} \frac{1}{T - 1} \sum_{t=2}^{T} \|x_t^{(i)} - \text{Model}_T(x_{t-1}^{(i)})\|^2. \tag{13}$$

Resuls are reported in Table 2 as averages accross the five splits, along with the standard deviation of the means, i.e.,

$$\bar{\varepsilon} = \frac{1}{5} \sum_{k=1}^5 \varepsilon_k \ , \ \hat{\sigma}_{\bar{\varepsilon}} = \frac{\sqrt{\frac{1}{5-1} \sum_{k=1}^5 (\varepsilon_k - \bar{\varepsilon})^2}}{\sqrt{5}}, \tag{14}$$

and

$$\bar{\xi} = \frac{1}{5} \sum_{k=1}^5 \xi_k \ , \ \hat{\sigma}_{\bar{\xi}} = \frac{\sqrt{\frac{1}{5-1} \sum_{k=1}^5 (\xi_k - \bar{\xi})^2}}{\sqrt{5}}. \tag{15}$$

Datasets are detailed below.

**Single Attractor.**  This system, studied in depth in Section 4.1 and also considered in Section 4.2, is a simple 2D ODE whose dynamic converges to a known attracting manifold [Brunton et al., 2016, 2022]. Trajectories are simulated via `odeint` from SciPy with 100 time steps, equally spaced with step size $\Delta t = 0.01$, using initial conditions sampled uniformly from the domain $[-2, 2]^2$. The continuous-time eigenvalues reported in Section 4.1 are obtained from the estimated discrete-time ones via the transformation $\lambda = \log(\lambda_{\text{disc}})/\Delta t$. The ground-truth Koopman eigenfunctions for this system are given by $\varphi_1 = x_2 - \frac{\lambda_1^*}{\lambda_1^* - 2\lambda_2^*} \cdot x_1^2$, $\varphi_2 = x_1$, and $\varphi_3 = x_1^2$.

**Duffing Oscillator.**  This system appears in the study of nonlinear oscillations and serves as a classical benchmark for testing methods in dynamical systems. More specifically, the dynamics follow:

$$\ddot{u} = -\delta \dot{u} - u(\beta + \mu u^2), \tag{16}$$

where the parameters $\delta$, $\beta$, and $\mu$ represent the damping coefficient, the linear stiffness, and the nonlinear stiffness respectively. Here, we choose the parameters $\delta = 0.5$, $\beta = -1$, and $\mu = 1$, following the setting used in [Williams et al., 2015, Otto and Rowley, 2019]. This nonlinear system exhibits two stable spirals at $u = \pm 1, \dot{u} = 0$ and a saddle at the origin. Trajectories are simulated using `odeint`, each consisting of 100 time steps with a fixed interval $\Delta t = 0.025$. Initial conditions are sampled uniformly from the domain $[-2, 2]^2$.

**Van der Pol Oscillator.**  The Van der Pol oscillator is a classical nonlinear system that exhibits self-sustained oscillations with a stable limit cycle. Its dynamics are governed by the second-order differential equation:

$$\dot{u} = v, \tag{17}$$
$$\dot{v} = \mu(1 - u^2)v - u, \tag{18}$$

where $\mu > 0$ controls the nonlinearity and damping strength. We follow the setup in Sinha et al. [2019] and set $\mu = 0.2$. Trajectories are simulated using `odeint`, with 100 time steps and a fixed step size of $\Delta t = 0.01$. Initial conditions are sampled uniformly from the square domain $[-4, 4]^2$.

**Lorenz System.** The Lorenz system is a classical example of a chaotic dynamical system, originally developed to model atmospheric convection. Its dynamics are governed by the following system of nonlinear differential equations:

$$\dot{u} = \sigma(v - u), \tag{19}$$
$$\dot{v} = u(\rho - w) - v, \tag{20}$$
$$\dot{w} = uv - \beta w, \tag{21}$$

where we have chosen $\sigma = 10$, $\rho = 28$, and $\beta = 8/3$, a commonly studied parameter regime known to induce chaotic behavior which was also considered in [Kostic et al., 2022]. We simulate the trajectories using RK45, with a time step $\Delta t = 0.01$ over 500 steps. Initial conditions are sampled uniformly from the cube $[-10, 10]^3$ centered at the origin.

### B.1.2 Real data

We also benchmark on three real datasets. Since the models are now evaluated on a single trajectory, we are only interested on online prediction error. Let the online prediction error at time step $t$ be $err_t = \|x_t - \text{Model}_{t-1}(x_{t-1})\|^2$. We report in Table 2 the temporal mean of $err_t$

$$\overline{err} = \frac{1}{T - t_0} \sum_{t=t_0+1}^{T} err_t, \tag{22}$$

and the standard deviation of $err_t$

$$\hat{\sigma}_{err} = \sqrt{\frac{1}{T - t_0 - 1} \sum_{t=t_0+1}^{T} (err_t - \overline{err})^2}. \tag{23}$$

**ETD** ⋆ We test on ETTh1 which is part of the Electricity Transformer Dataset (ETDataset)[0] introduced by Zhou et al. [2021]. ETTh1 contains data recorded at hourly intervals for roughly two years from an electricity transformer station. The dataset consists of a single trajectory of 6 features: HUFL (High Use Frequency Load), HULL (High Use Low Load), MUFL (Medium Use Frequency Load), MULL (Medium Use Low Load), LUFL (Low Use Frequency Load), and LULL (Low Use Low Load). In our setting, we retain 200 time steps and aim to learn the transformer's load profile as a dynamical system.

**EEG** ⋆ To assess the COLoKe's performance on high-dimensional data, we evaluate it on electroencephalogram (EEG) recordings with 64 channels from the PhysioNet EEG Motor Movement/Imagery dataset [1]. We use the recording of Subject 1 in the *eyes open* condition. The original signal consists of one minute of data sampled at 160 Hz. We downsample it to 16 Hz, resulting in a trajectory of dimension $d = 64$ and of length $T = 976$.

**CASES-99** ⋆ To further evaluate COLoKe on real-world atmospheric turbulence, we use data from the *CASES-99*[2] (Cooperative Atmospheric Surface Exchange Study 1999) field experiment. The CASES-99 examined boundary-layer turbulence using high-frequency measurements from instrumented towers and remote sensing systems. In our setting, we use the recordings from the 55m tower. Specifically, we extract the three-dimensional wind components $u$, $v$, and $w$ and retain 500 time steps. This yields a trajectory of dimension $d = 3$ and length $T = 500$.

*Remark* B.1. For all datasets, data prior to time $t_0$ are used to initialize the model parameters, including those of the Conformal PI control for COLoKe [Angelopoulos et al., 2023]. In the case of synthetic datasets, we set $t_0 = T/5$, resulting in $t_0 = 20$ for the single attractor, Duffing oscillator, and Van der Pol oscillator, and $t_0 = 100$ for the Lorenz system. For all real datasets, we use $t_0 = 100$.

---

[0] https://github.com/zhouhaoyi/ETDataset
[1] https://physionet.org/content/eegmmidb/1.0.0/
[2] https://www.eol.ucar.edu/field_projects/cases-99

## B.2 Baseline Models

This section details the baseline models used in our experiments, including their initialization, neural network architectures (when applicable), and online training procedures.

Among the various models considered, the most commonly used in the literature are ODMD and our variant OEDMD (also known as RR-EDMD), which we briefly recall below.

**ODMD.** Online Dynamic Mode Decomposition [Zhang et al., 2019] incrementally estimates a linear model from sequential data, enabling efficient updates without storing the entire dataset. Given a trajectory $\{x_1, x_2, \ldots x_{t_0}\}$ available to time $t_0$ for model initialization. Let $X_{t_0} = [x_1, x_2, \ldots, x_{t_0-1}]$ and $Y_{t_0} = [x_2, x_3, \ldots, x_{t_0}]$. The matrix $K_{t_0}$ is computed by

$$K_{t_0} = Y_{t_0} X_{t_0}^+ = Y_{t_0} X_{t_0}^\top (X_{t_0} X_{t_0}^\top)^{-1}. \tag{24}$$

Let

$$Q_{t_0} = Y_{t_0} X_{t_0}^\top, \text{ and } P_{t_0} = \left( X_{t_0} X_{t_0}^\top \right)^{-1}, \tag{25}$$

then $K_{t_0} = Q_{t_0} P_{t_0}$. We start the online learning when a new observation $x_t$ for $t = t_0 + 1$ arrives:

$$Q_t = Q_{t-1} + x_t x_{t-1}^\top, \text{ and } P_t^{-1} = P_{t-1}^{-1} + x_{t-1} x_{t-1}^\top. \tag{26}$$

The matrix $K_t = Q_t P_t$ is obtained by inverting $P_t^{-1}$ using Sherman Morrison formula

$$P_t = \left( P_{t-1}^{-1} + x_{t-1} x_{t-1}^\top \right)^{-1} = P_{t-1} - \frac{P_{t-1} x_{t-1} x_{t-1}^\top P_{t-1}}{1 + x_{t-1}^\top P_{t-1} x_{t-1}} \tag{27}$$

This online formulation allows for recursive updates of $K_t$ and yields an efficient and memory-light algorithm for learning linear dynamics in real time. The readers are referred to [Zhang et al., 2019] for a comprehensive explication of ODMD.

**OEDMD.** We first present Recursive EDMD [Sinha et al., 2019] and point out its limitations for pure online setting, in order to introduce the variant OEDMD. Choose a fixed dictionary $\Phi : \mathcal{X} \to \mathbb{R}^r$

$$\Phi(x) := [\phi_1(x), \phi_2(x), \ldots, \phi_r(x)]^\top. \tag{28}$$

Let $X_{t_0} = [\Phi(x_1), \Phi(x_2), \ldots, \Phi(x_{t_0-1})]$ and $Y_{t_0} = [\Phi(x_2), \Phi(x_3), \ldots, \Phi(x_{t_0})]$, then the initialization of R-EDMD follows the equations (24) and (25). The recursive updates of $K_t$ are given by equations (26) and (27). To get predictions in the state space $\mathcal{X}$, one should solve a least squares problem

$$\min_C \sum_{k=1}^t \|x_k - C\Phi(x_k)\|^2. \tag{29}$$

Typically, this involves the storage of all historical data. Therefore, we propose to calculate the linear reconstruction matrix $C$ using the current buffer $\mathcal{D}_t$:

$$\min_C \sum_{k=t-w}^t \|x_k - C\Phi(x_k)\|^2. \tag{30}$$

Since the modified problem may not have a closed form solution due to the buffer size, we use regularized pseudo-inverse to efficiently update $C$. Rewrite the problem with Ridge regularization ($\rho = 10^{-6}$ for all experiments)

$$\min_C \sum_{k=t-w}^t \|x_k - C\Phi(x_k)\|^2 + \rho\|C\|_F^2. \tag{31}$$

Let $Z = [x_{t-w}, \ldots, x_t]$ and $\Phi_Z = [\Phi(x_{t-w}), \ldots, \Phi(x_t)]$, then we have the closed form solution

$$C = Z\Phi_Z^\top \left( \Phi_Z \Phi_Z^\top + \rho I \right)^{-1}. \tag{32}$$

In the original work [Sinha et al., 2019], the authors used Radial Basis Functions (RBF) as the fixed dictionary. However, to estimate informative centers for RBF, one needs to have access to the

full trajectory up to time $T$. When estimating the centers with trajectory up to time $t_0$, the model gives poor results on all datasets for both metrics. Therefore, we choose a polynomial dictionary of degree 2, which provides a lifted representation dimension comparable to other baseline models. Increasing the degree beyond 2 offers no significant performance gain. On the contrary, it degrades the performance on the real dataset and results in substantially higher computational costs.

We now describe the deep models used in our experiments. All models are trained using the `AdamW` optimizer with a learning rate of $10^{-3}$, both during parameter initialization and online training. The initialization phase consists of 4000 epochs for synthetic datasets and 5000 epochs for the real-world dataset. The neural network architectures used in each model are detailed below.

**COLoKe.** The detailed presentation of COLoKe can be found in Section 3. The neural network $\tilde{\Phi}$ is fully connected with architecture $\{d, 32, 16, 8, m-d\}$ for synthetic datasets and $\{d, 64, 32, 16, m-d\}$ for the real dataset. The dimension of lifted representation $m$ is chosen to be $d + \lceil d/2 \rceil$ for all experiments. Model parameters and initial conformity threshold are initialized with $\{x_0, \cdots, x_{t_0}\}$ as already discussed. The initial threshold $q_{t_0+1}$ is set to be $1 - \alpha$ quantile of the set of scores $\{s_{w+1}, \ldots, s_{t_0}\}$ computed with the initialized model. The model parameters $(\theta_t, K_t)$ are updated online according to Algorithm 1. For all synthetic datasets, the hyperparameters for Conformal PI procedure are $\alpha = 0.5, \gamma = 0.1, C_{sat} = 5$, and we set $C_{sat} = 10$ for the real dataset. Note that the coverage guarantees [Angelopoulos et al., 2023] hold for any value of $\gamma > 0$. We set $\gamma = 0.1$ in our experiments as in [Angelopoulos et al., 2023], although one may implement adaptive strategies as in [Bhatnagar et al., 2023]. The choice of $\alpha$ should reflect the trade-off between computational efficiency and predictive accuracy, depending on the requirements of the application. When $\alpha \approx 1$, accuracy is prioritized over speed, whereas for $\alpha \ll 1$, computational speed is more important. In our experiments, we selected $\alpha = 0.5$ as a balanced compromise between these two objectives.

**OLoKe.** The only difference between COLoKe and OLoKe is the online training strategy. For every new buffer $\mathcal{D}_t$, OLoKe performs a fixed number of iterations.

**OnlineAE.** We implement the model in [Liang et al., 2022]. The original work tackles the problem of Model Predictive Control (MPC). We aim only to perform online learning of dynamics. For synthetic datasets, the encoder architecture is $\{d, 32, 16, 8, m\}$ and the decoder architecture is $\{m, 8, 16, 32, d\}$. For the real dataset, the encoder architecture is $\{d, 64, 32, 16, m\}$ and the decoder architecture is $\{m, 64, 32, 16, d\}$. The dimension of lifted representation m is chosen to be $d + \lceil d/2 \rceil$. Thus, the model architecture of OnlineAE aligns with COLoKe and OLoKe. The loss function at time $t$ is defined as

$$\mathcal{L}_t(\mathcal{D}_t, \Phi_t, \Psi_t, K_t) = \sum_{k=t-w}^{t} \underbrace{\|\Psi_t\left[K_t\Phi_t(x_{k-1})\right] - x_k\|^2}_{\text{prediction loss}} + \underbrace{\|\Psi_t \circ \Phi_t(x_k) - x_k\|^2}_{\text{autoencoding loss}}$$
$$+ \underbrace{\|K_t\Phi_t(x_{k-1}) - \Phi_t(x_k)\|^2}_{\text{lifted prediction loss}},$$

where $\Phi_t$ is the encoder and $\Psi_t$ is the decoder. The online training strategy consists of fixed iterations with $N_{\text{iter}} = 100$ for synthetic datasets and $N_{\text{iter}} = 500$ for the real dataset, which aligns with OLoKe.

## C  Adaptive update behaviors of COLoKe

We provide supplementary quantitative and qualitative analyses concerning the update frequency of COLoKe. Specifically, we report key summary statistics (see Table 3) that characterize its update behavior and relate it to properties of each dynamical system (see Appendix B.1). These statistics include:

1. the number and frequency of update triggers
2. the average interval between two consecutive triggered updates
3. the maximum interval between two consecutive triggered updates.

Table 3: Update statistics.

| Dataset | Total steps | Triggers | Percentage | Avg. interval | Max interval |
|---------|-------------|----------|------------|---------------|--------------|
| Single attractor | 80 | 37 | 0.46 | 2.16 | 8 |
| Duffing oscillator | 80 | 38 | 0.48 | 2.08 | 6 |
| VdP oscillator | 80 | 40 | 0.50 | 2.00 | 9 |
| Lorenz system | 400 | 192 | 0.48 | 2.07 | 10 |

Overall, the proportion of steps triggering updates remains consistently close to 48%, but the temporal structure of these updates varies appreciably across systems. This variation illustrates how the conformal triggering mechanism adapts to the underlying stability and complexity of each dynamical regime. In summary:

- **Single attractor:** More iterations are needed initially, but once near the attractor, the model requires fewer corrections.

- **Duffing oscillator:** Updates are more uniformly distributed, consistent with moderately chaotic behavior and frequent small deviations.

- **VdP oscillator:** Updates tend to cluster, with longer stretches of stability that align with the smoother, low-variability regimes of this system.

- **Lorenz system:** Many updates occur early on due to the highly unstable transient phase, followed by sparser updates as the model adapts.

These behaviors emerge naturally from our conformal-triggering mechanism, with no dataset-specific tuning or heuristics required, highlighting its adaptability across diverse dynamical settings.

## D    Computation time comparison

To complement the results reported in Table 2, we provide a full comparison of execution time (in seconds) across all datasets. Table 4 shows that COLoKe consistently outperforms neural network baselines—OnlineAE and OLoKe—often by a substantial margin. As expected, the purely linear ODMD remains the fastest method overall, but it lacks the capacity to learn meaningful nonlinear embeddings. OEDMD can approximate nonlinear observables, but it relies on a predefined dictionary, which limits its expressivity and scalability. As a result, these methods underperform compared to COLoKe (cf. Table 2). Despite its reduced runtime, achieved through the conformal-triggering mechanism which prevents unnecessary updates, COLoKe maintains *best predictive accuracy*, highlighting its favorable cost–performance trade-off.

Table 4: Computation time (in seconds) of each method across all datasets.

| Dataset | ODMD | OEDMD | OnlineAE | OLoKe | COLoKe |
|---------|------|-------|----------|-------|--------|
| Single attractor | 1.01 | 42.25 | 27.08 | 19.25 | 11.50 |
| Duffing oscillator | 1.03 | 43.36 | 26.77 | 19.23 | 9.72 |
| VdP oscillator | 1.04 | 44.06 | 27.00 | 19.60 | 10.94 |
| Lorenz system | 5.42 | 470.53 | 130.16 | 266.25 | 107.96 |
| ETD $\star$ | 0.10 | 0.52 | 31.46 | 22.25 | 13.31 |
| EEG $\star$ | 0.03 | 0.50 | 4.94 | 16.55 | 3.67 |
| Turbulence $\star$ | 0.01 | 0.22 | 22.83 | 16.15 | 11.43 |

