# OpenReview forum: "Conformal Online Learning of Deep Koopman Linear Embeddings"
_NeurIPS.cc/2025/Conference — NeurIPS 2025 poster_

### Official Review · Reviewer_vi1N · 2025-06-27

**Clarity:** 3
**Significance:** 2
**Originality:** 2
**Rating:** 4
**Confidence:** 5

**Summary:**

This paper introduces an event triggering mechanism based on conformal prediction to control the frequency of model updates to achieve an adaptive online Koopman embedding learning. The proposed method performs well on 4 synthetic datasets. However, the introduction of event triggering mechanism in online learning methods lacks novelty. The performance of the proposed method on a real dataset is also marginal and lacks validation on more real datasets. And the comparison methods are not enough and outdated.

**Questions:**

N.A.

**Ethical Concerns:**

["NO or VERY MINOR ethics concerns only"]

**Final Justification:**

I'd like to thank the authors for their response. I've raised the overall score.

**Limitations:**

See weaknesses above.

**Paper Formatting Concerns:**

N.A.

**Quality:**

2

**Strengths And Weaknesses:**

Strengths:

1.	Introduce conformal online learning as an event triggering mechanism to form an adaptive online Koopman embedding learning method to reduce the frequency of model updates.
2.	The proposed method performs well on 4 synthetic datasets.

Weaknesses:

1.	The idea of the proposed conformal online learning for controlling the frequency of model updates is quite similar to the event triggering mechanisms and some online learning methods [1] have proposed similar frameworks to control the frequency of model updates, hence it is not novel.

[1] Guan, M., Wen, C., Shan, M., Ng, C. L., & Zou, Y. (2018). Real-time event-triggered object tracking in the presence of model drift and occlusion. IEEE Transactions on Industrial Electronics, 66(3), 2054-2065.

2.	From the tables, the improvement of the proposed method on the real dataset is marginal and even worse than some comparison methods.

3.	Only one real dataset is used, which is not enough.

4.	Authors claimed that the introduction of conformal prediction to guide adaptive model updates will significantly improve speed and unlock practical Koopman learning in real-time. But there is no speed comparison between the proposed method and other online Koopman embedding learning methods.

5.	To show the improvement of the proposed method clearer, introducing some visualization will be better, for example, introducing visualizations of predicted state space trajectories for each system using different methods and MSE curves during training.

6.	The comparison methods are not enough and outdated. Although authors listed several comparison methods in Table 1, only ODMD (2019) and OnlineAE (2022) are used in comparison which are quite outdated and the superiority of the proposed method over these methods is not convincing. Some real-time advanced adaptive Koopman embedding learning methods [2,3] are required to compare.

[2] Singh, R., Sah, C. K., & Keshavan, J. (2024). Adaptive Koopman embedding for robust control of complex nonlinear dynamical systems. arXiv preprint arXiv:2405.09101.

[3] Li, B., Ma, Y., Kutz, J. N., & Yang, X. (2023). The adaptive spectral Koopman method for dynamical systems. SIAM Journal on Applied Dynamical Systems, 22(3), 1523-1551.

---

> ### Author Rebuttal · Authors · 2025-07-30
>
> We would like to thank the reviewer for their detailed and thoughtful feedback. Nevertheless, we respectfully believe that the current rating does not adequately reflect the strength of our contribution since our method is 1) theoretically grounded, 2) empirically validated on multiple benchmark datasets commonly studied in the Koopman learning literature and 3) fully reproducible. We have also carried out new experiments to directly address the reviewer’s concerns, including the addition of two real-world datasets (see W3) and a detailed comparison of execution times (see W4). We hope that our clarifications (see W1, W2 and W6) and additional results provide a more complete picture of our contributions and will convince the reviewer to reconsider their evaluation. If there are additional concerns that contributed to the negative score, we would gladly welcome the opportunity to address them as well.
>
> - [W1] *"The idea of the proposed conformal online learning [...] is not novel."*
>
>     We respectfully disagree. While event-triggered mechanisms themselves are not new, the novelty of our contribution lies in introducing conformal prediction as a **principled and statistically grounded approach** to triggering updates in online learning. Unlike works such as [1], which rely on fixed or heuristic thresholds (see their Eq. 20), our method leverages **adaptive thresholdings derived from conformal scores in a bespoke manner**. This provides coverage guarantees and enforces statistical significance into the triggering process. To the best of our knowledge, this is the first work to leverage the power and flexibility of conformal prediction in the context of event-triggered online learning.
>
> - [W2] *"From the tables, the improvement of the proposed method on the real dataset is marginal and even worse than some comparison methods."*
>
>     This may be a misinterpretation of the results.**Our method achieves the best online error**, which is the primary performance metric in online learning. We believe that the results in the table faithfully reflect the model’s ability to make accurate predictions during learning. For completeness, we also report the generalization error (i.e., post-hoc performance on new trajectories), where COLoKe remains competitive and exhibits significantly lower variance than state-of-the-art methods. This robustness further supports the practical reliability of our approach, especially for real-world datasets where classical assumptions in Koopman operator learning (e.g., autonomous system, absence of distribution drift) no longer hold.
>
> - [W3] *"Only one real dataset is used, which is not enough."*
>
>     We thank the reviewer for this helpful suggestion and have extended our numerical study with **two additional experiments on real-world datasets**: (i) EEG recordings with 64 channels from the PhysioNet EEG Motor Movement/Imagery dataset [Dataset 1], and (ii) turbulence measurements from the CASES-99 atmospheric field experiment [Dataset 2]. The table below reports the online errors, with standard deviations shown in parentheses. These results confirm that our method performs reliably in real-world challenging scenarios. We will include these experiments in the revised version of the manuscript.
>
>     | Dataset       | COLoKe         | ODMD           | OnlineAE       | OLoKe          | OEDMD          |
>     |---------------|----------------|----------------|----------------|----------------|----------------|
>     | EEG (x1e-3)  | **7.9 (8.5)** | 8.3 (11.3) | 12.4 (18.1) | 9.3 (10.4) | 8.0 (11.1) |
>     | Turbulence (x1e-4)   | **4.7 (5.9)** | 5.5 (7.5) | 6.4 (8.6) | 5.5 (6.1) | 7.3 (8.9) |
>
>     We would also like to note that real-world datasets are still rarely used in the Koopman learning literature, where controlled and well-understood systems are typically favored to analyze the spectral properties of the learned Koopman operator. We welcome the opportunity, prompted by your remark, to help bridge this gap and demonstrate how our method can robustly extend to realistic, high-dimensional settings beyond traditional benchmarks.
>
> - [W4] *"[...] But there is no speed comparison between the proposed method and other online Koopman embedding learning methods."*
>
>     We thank the reviewer for this useful observation. While our original analysis provided limited timing insights (Fig. 2b and, to some extent, appendix C2), we now include a full comparison of execution times (in seconds) across all datasets to complement Table. 2:
>
>
>     | Dataset       | ODMD         | OEDMD            | OnlineAE       | OLoKe          | COLoKe         |
>     |---------------|----------------|----------------|----------------|----------------|----------------|
>     | Single attractor  | 1.01 | 42.25 | 27.08 | 19.25 | 11.50 |
>     | Duffing oscillator   | 1.03 | 43.36 | 26.77 | 19.23 | 9.72 |
>     | VdP oscillator   | 1.04 | 44.06 | 27.00 | 19.60 | 10.94 |
>     | Lorenz system   | 5.42 | 470.53 | 130.16  | 266.25  | 107.96 |
>     | ETD   | 0.10 | 0.52 | 31.46 | 22.25  | 13.31  |
>
>     These results show that **COLoKe consistently outperforms nonlinear baselines** in speed, often by a wide margin, while maintaining superior predictive accuracy (cf. Table 2). Only the purely linear ODMD is faster, yet unable to learn expressive nonlinear embeddings.
>
>     To further illustrate COLoKe’s efficiency, we compared it to OLoKe trained with different fixed iteration budgets on the real high-dimensional EEG dataset [Dataset 1]:
>
>     |        | COLoKe  | OLoKe 5 | OLoKe 10 | OLoKe 50 |OLoKe 100 | OLoKe 500 |
>     |---------------|----------------|----------------|----------------|----------------|----------------|----------------|
>     | Online error (x1e-3)  | **7.9 (8.5)** | 9.4 (10.4) | 8.8 (9.5) | 8.1 (9.5) | 8.2 (9.6)  | 8.3 (8.8) |
>     | Execution time (s)   | **2.98** | 11.12 | 15.29 | 50.44 | 92.81 | 444.84 |
>
>     This confirms that COLoKe not only avoids unnecessary computation through adaptive updates but also yields the best accuracy–speed trade-off without manual tuning. Its data-driven update strategy offers a principled and efficient alternative to fixed-iteration baselines. In the revised manuscript, **we will include Pareto fronts visualizations** (online error vs. execution time) to clearly illustrate the trade-offs between accuracy and speed across all methods and datasets.
>
> - [W5] *"To show [...] predicted state space trajectories for each system using different methods and MSE curves during training."*
>
>     We thank the reviewer for this valuable suggestion. We will include such illustrations in the final version. Unfortunately, we are unable to share them during the rebuttal phase, as including images is not permitted.
>
> - [W6] *"The comparison methods are not enough and outdated. [...] Some real-time advanced adaptive Koopman embedding learning methods [2,3] are required to compare."*
>
>     We thank the reviewer for this comment and would like to clarify the discussion about related works in two points.
>     - First, mentioning [3] which specifically focuses on the numerical integration of ODEs using the Koopman framework is a bit surprising to us, because pertaining to a different type of application. As for [2] and similar adaptive Koopman embedding methods, they operate in a fundamentally different setting: **full trajectories are available ahead of time and these full trajectories are used to fit the model**. Additionally, their adaptive component is designed specifically for model predictive control (MPC). In contrast, our method performs **online Koopman learning from streaming data**, with no access to future states at any point. The comparison is therefore not just unfair, it concerns a completely different problem formulation, both in terms of assumptions and objectives. In the case where the reviewer would find it helpful, we would be happy to add a discussion of MPC in the paper's introduction.
>
>     - Second, to the best of our knowledge, the only works related to our problem are the ones listed in Table 1. We have not included some of them in our experiments, since they cannot fulfill all the conditions of a purely online learning setting (see the "Baselines" paragraph in Section 4.2). Please also note that OEDMD appearing in our experiments is just a reimplementation of R-EDMD [Sinha et al., 2023]. While DMD, EDMD and their online variants may appear dated, they remain widely used due to their strong theoretical foundations and performance guarantees, which serve as reliable baselines withstanding the test of time.
>
> References:
>
> - [Dataset 1] https://physionet.org/content/eegmmidb/1.0.0/
> - [Dataset 2] https://www.eol.ucar.edu/field_projects/cases-99

---

### Official Review · Reviewer_vz6L · 2025-07-03

**Clarity:** 3
**Significance:** 3
**Originality:** 3
**Rating:** 5
**Confidence:** 2

**Summary:**

This paper introduces COLoKe (Conformal Online Learning of Koopman embeddings), a framework for learning deep Koopman models for nonlinear dynamical systems in an online/streaming data setting. The method combines the learning of deep Koopman embeddings with an adaptive update mechanism inspired by conformal prediction (CP). Instead of using conformal methods for uncertainty quantification, COLoKe leveragesthe conformity score (error from a multi-step prediction) to measure how consistent the incoming data is with the current Koopman model. A gradient-based refinement of the model is performed only if this score exceeds a threshold. The primary advantage of COLoKe is that it reduces computational cost and prevents overfitting by avoiding unnecessary updates. The authors also provide a theoretical analysis establishing a dynamic regret bound under certain assumptions. Evaluations on several benchmark dynamical systems, demonstrate that COLoKe can achieve high prediction accuracy while requiring fewer updates than non-adaptive or offline methods.

**Questions:**

1. Can you provide some intuition why $\sum q_t$ should grow sub-linearly? Under what real-world conditions would it fail, e.g., non-stationarity, turbulence, drift?

1. Can you comment regarding the scalability of the COLoKe framework to higher dimensions (e.g. 2D/3D Navier-Stokes systems, or other complex systems exhibiting chaotic and turbulent behavior)?

**Ethical Concerns:**

["NO or VERY MINOR ethics concerns only"]

**Final Justification:**

This work presents the method COLoKe which is a technically sound and well-motivated method for adaptive updates. Strong empirical evaluation validates the method. I believe it is significant contribution to the community.

**Limitations:**

yes

**Quality:**

3

**Strengths And Weaknesses:**

*Strengths*
- The paper is technically well-motivated; reducing unnecessary updates of the Koopman bases is a clear goal, and the approach using conformal prediction is quite plausible. The technique to preventing unnecessary updates is repurposing an existing well-established method rather than reinventing the wheel.
- The empirical evaluation is a key strength. COLoKe shows strong performance across several benchmarks (Table 2), often outperforming baselines. Crucially, it outperforms OLoKe (an ablation with fixed updates), supporting the main claim that adaptive updates are beneficial.

*Weaknesses*
- The performance of the COLoKe method may be highly dependent on several hyperparameters (e.g. alpha), but the paper lacks discussion or guidance how to go about choosing such parameters.

---

> ### Author Rebuttal · Authors · 2025-07-30
>
> We would like to sincerely thank the reviewer for their insightful and constructive comments. We are grateful for the opportunity to further strengthen our work by adding these points, which we believe will significantly enrich the discussion around our contributions.
>
>
> - [W] *"The performance of the COLoKe method may be highly dependent on several hyperparameters (e.g. alpha), but the paper lacks discussion or guidance how to go about choosing such parameters."*
>
>     We thank the reviewer for this valuable observation. To guide the choice of our two hyperparameters, we propose adding the following paragraphs after Eq. 3 and Eq. 9, respectively
>     - "Note that classical guarantees in conformal prediction hold for any value of $\gamma>0$. In our experiments, we will set $\gamma=0.1$ as in [Angelopoulos et al., 2023a] although one may implement adaptative strategies as in [Ref 1] and references therein."
>     - "The choice of $\alpha$ should reflect the trade-off between computational efficiency and prediction accuracy, depending on the requirements of the application. When $\alpha\approx 1$, accuracy is prioritized over speed, whereas for $\alpha\ll 1$, computational speed is more important. In our experiments, we selected $\alpha=0.5$ as a balanced compromise between these two objectives."
>
> - [Q1] *"Can you provide some intuition why $\sum q_t$ should grow sub-linearly? Under what real-world conditions would it fail, e.g., non-stationarity, turbulence, drift?"*
>
>     We thank the reviewer for their insightful question. This touches on a key research direction that we plan to explore in future work. To provide some intuition, consider a hypothetical scenario in which the cumulative sum grows linearly, this would imply that the $q_t$ remain roughly constant over time. From a conformal prediction perspective, such behavior suggests that the model's predictive uncertainty does not decrease, even as more observations are collected. In other words, the model fails to learn a representation that generalizes or improves with data. As rightfully pointed, such problematic scenario may correspond to cases involving non-autonomous systems or distribution shifts, or cases where the model is insufficiently trainable or expressive to capture complex dynamics, such as those encountered in turbulent regimes.
>
> - [Q2] *"scalability of the COLoKe framework to higher dimensions (e.g. 2D/3D Navier-Stokes systems, or other complex systems exhibiting chaotic and turbulent behavior"*
>
>     We thank the reviewer for this important question. To assess the scalability and robustness of COLoKe in higher-dimensional and complex settings, we conducted additional experiments on two real-world datasets: (i) EEG recordings with 64 channels from the PhysioNet EEG Motor Movement/Imagery dataset [Dataset 1], and (ii) real-world turbulence measurements from the CASES-99 atmospheric field experiment [Dataset 2]. Both datasets are governed by complex dynamics and are inherently noisy, making them well-suited to evaluate the practical utility of our method. The table below reports the online errors, with standard deviations shown in parentheses. These results demonstrate that our approach generalizes well beyond low-dimensional synthetic systems and remains effective in realistic high-dimensional scenarios. We will include these findings in the revised version of the manuscript.
>
>     | Dataset       | COLoKe         | ODMD           | OnlineAE       | OLoKe          | OEDMD          |
>     |---------------|----------------|----------------|----------------|----------------|----------------|
>     | EEG (x1e-3)  | **7.9 (8.5)** | 8.3 (11.3) | 12.4 (18.1) | 9.3 (10.4) | 8.0 (11.1) |
>     | Turbulence (x1e-4)   | **4.7 (5.9)** | 5.5 (7.5) | 6.4 (8.6) | 5.5 (6.1) | 7.3 (8.9) |
>
> References:
>
> - [Ref.1] Bhatnagar, A., et al. Improved online conformal prediction via strongly adaptive online learning. In ICML 2023.
> - [Dataset 1] https://physionet.org/content/eegmmidb/1.0.0/
> - [Dataset 2] https://www.eol.ucar.edu/field_projects/cases-99

---

> > ### Author Response · Authors · 2025-08-08
> >
> > We would like to thank the reviewer again for their thorough evaluation of our work and for their questions. We would be grateful to know if our additional experiments and clarifications have addressed your concerns ?

---

> > > ### Comment · Area_Chair_fJ5i · 2025-08-08
> > >
> > > Dear reviewer vz6L,
> > >
> > > Can you please take a moment to acknowledge the authors' rebuttal?
> > >
> > > I know that your assessment was positive to begin with, but additionally, I would appreciate if you can add a small note stating on which, if any, points has your assessment changed given the authors' responses.

---

> > ### Comment · Reviewer_vz6L · 2025-08-08
> >
> > Thank you to the authors for including the additional discussions in the paper as well as conducting the additional experiments. The authors have fully addressed my questions. I will keep my current positive score.

---

### Official Review · Reviewer_4zQ8 · 2025-07-03

**Clarity:** 4
**Significance:** 3
**Originality:** 3
**Rating:** 5
**Confidence:** 3

**Summary:**

The authors study the problem of approximating nonlinear dynamical systems using linear Koopman operator theory. Specifically, they focus on the online learning setting, where the goal is to continuously learn a finite-dimensional, linear representation of the dynamics from streaming data.

Unlike traditional approaches that use fixed dictionaries of observables, the authors propose to learn the Koopman-invariant embedding using a neural network, which maps the state into a latent space where the dynamics evolve approximately linearly. This deep Koopman embedding includes both a nonlinear component (learned) and the original state itself, ensuring partial interpretability and reconstruction capability.

To address the challenges of overfitting and computational inefficiency in online learning, the authors introduce a conformal prediction-based update strategy. Instead of updating the model at every time step, they compute a prediction conformity score based on multi-step prediction errors in the lifted space. Updates are triggered only when this score exceeds a dynamically calibrated threshold.

This threshold is updated using a Proportional-Integral (PI) controller, inspired by control theory, which tracks the frequency of conformity violations and adjusts sensitivity accordingly. This mechanism ensures that the model adapts only when necessary.

**Questions:**

Can you make additional analysis on how frequently updates had to be done for table 2.
Updates are triggered only when this score exceeds a dynamically calibrated threshold.
I would suggest to make time series visualisation with trajectories of true dynamics, predictions, and vertical lines that indicate when updates to the model had to be done.

Can you try simple benchmark of reservoir computing for online learning and compare it to your approach?

**Ethical Concerns:**

["NO or VERY MINOR ethics concerns only"]

**Final Justification:**

I would stay with my rating of accept. Authors have done additional work in rebuttal phase, that showed me that accept is a good decision.
Overall good written paper, good baselines, and solid theoretical framework with applications.

**Limitations:**

Author clearly state limitations in their paper.

**Quality:**

3

**Strengths And Weaknesses:**

Strengths:
- simple framework rooted in strong theory
- very applied

Weaknesses:
- no big problems

---

> ### Author Rebuttal · Authors · 2025-07-30
>
> We would like to sincerely thank the reviewer for their kind evaluation. Their valuable suggestions will greatly improve the quality of our paper. We will make sure to include them in the revised version of the manuscript.
>
> - [Q1] *"Can you make additional analysis on how frequently updates had to be done for table 2. Updates are triggered only when this score exceeds a dynamically calibrated threshold. I would suggest to make time series visualisation with trajectories of true dynamics, predictions, and vertical lines that indicate when updates to the model had to be done."*
>
>     We thank the reviewer for these great suggestions. Unfortunately, we are unable to share such illustrations during the rebuttal phase, as including images is not permitted. We will include them in the camera-ready version.
>     Nevertheless, we provide below additional quantitative and qualitative analyses on the frequency of model updates. While plots would offer more visual insights, we instead report key statistics summarizing the update behavior. Across all datasets in Table 2, we report: (i) the total number of steps, (ii), the number of update triggers, (ii) the percentage of time steps at which a trigger occurred, as well as the average (iv) and largest (v) time steps between two consecutive triggered updates.
>
>     | Dataset       | Total steps         | Triggers           | Percentage   | Avg interval        | Max interval         |
>     |---------------|----------------|----------------|----------------|----------------|----------------|
>     | Single attractor   |  80  | 37  | 0.46 | 2.16 | 8  |
>     | Duffing oscillator |  80  | 38  | 0.48 | 2.08 | 6  |
>     | VdP oscillator     |  80  | 40  | 0.50 | 2.00 | 9  |
>     | Lorenz system      | 400  | 192 | 0.48 | 2.07 | 10 |
>     | ETD                | 100  | 49  | 0.49 | 2.02 | 6  |
>
>     While the overall frequency of updates is consistently around 48\%, the timing and distribution of those updates vary significantly across systems. This variation reflects how the conformal triggering mechanism adapts to the complexity and stability of each system’s dynamics. In short:
>
>     - **VdP oscillator and ETD**: Updates are clustered, with long intervals of stability matching the regular, low-variability regimes of these systems.
>     - **Duffing oscillator**: Updates are more uniformly distributed, reflecting moderate chaotic behavior with frequent small deviations.
>     - **Lorenz system**: Frequent early updates reflect the chaotic and unstable initial phase; they become sparser as the model adapts.
>     - **Single Attractor**:  Initial updates require more iterations, but once near the attractor, the model requires fewer corrections.
>
>     We will include a more detailed discussion in the revised manuscript to highlight how each dataset’s specific dynamics influence the update behavior. Note that these update patterns emerge naturally from our conformal-triggering mechanism, without requiring any dataset-specific tuning or heuristics, highlighting its adaptability across different dynamical regimes.
>
>
> - [Q2] *"Can you try simple benchmark of reservoir computing for online learning and compare it to your approach?"*
>
>     We thank the reviewer for this suggestion. We implemented a benchmark using reservoir computing (RC) across all datasets in Table 2. We used the *reservoirpy* package [Ref. 1] and focused on Echo State Networks as the RC architecture. We performed a grid search on two key hyperparameters: (i) the number of internal (reservoir) nodes chosen from the set \{10, 15, 20, 50, 100, 200, 500, 800, 1000, 2000, 3000, 4000, 5000\} and (ii) the learning rate of the Recursive Least Squares algorithm [Ref. 2], chosen from the set \{1e-3, 1e-4, 1e-5, 1e-6, 1e-7\}. For each dataset, we report the RC configuration that achieved (i) the lowest online error and (ii) the lowest generalization error. The first line gives the generalization error, and the second line the online error.
>
>     | Dataset           | RC (best online error)       | RC (best generalization) | COLoKe                 |
>     |-------------------|------------------------------|--------------------------------|------------------------------|
>     | Single attractor | 8.1 x 1e-2 (9.0 x 1e-3)          | 3.0 x 1e-3 (3.0 x 1e-4)             | **2.4 x 1e-7 (3.6 x 1e-8)**      |
>     |                   | 2.2 x 1e-5 (1.0 x 1e-5)          | 3.7 x 1e-4 (3.8 x 1e-4)             | **7.6 x 1e-7 (9.6 x 1e-8)**      |
>     |                   |                              |                                |                              |
>     | Duffing oscillator | 9.5 x 1e-2 (1.9 x 1e-3)          | 4.0 x 1e-3 (8.2 x 1e-5)             | **3.1 x 1e-6 (2.3 x 1e-7)**      |
>     |                   | 1.0 x 1e-4 (1.2 x 1e-5)          | 6.7 x 1e-4 (4.1 x 1e-4)             | **7.3 x 1e-5 (1.9 x 1e-5)**      |
>     |                   |                              |                                |                              |
>     | VdP oscillator | 3.9 x 1e-1 (5.3 x 1e-2)          | 3.0 x 1e-2 (4.6 x 1e-4)             | **3.8 x 1e-4 (1.2 x 1e-5)**      |
>     |                   | **9.0 x 1e-5 (3.5 x 1e-5)**          | 3.5 x 1e-3 (2.8 x 1e-3)             | 6.0 x 1e-4 (1.4 x 1e-4)      |
>     |                   |                              |                                |                              |
>     | Lorenz system  | 2.8 x 1e1 (2.3 x 1e0)           | 7.3 x 1e0 (9.1 x 1e-1)              | **6.5 x 1e-3 (1.0 x 1e-4)**      |
>     |                   | 2.6 x 1e-2 (2.2 x 1e-2)          | 4.3 x 1e0 (3.4 x 1e0)               | **3.3 x 1e-3 (1.1 x 1e-4)**      |
>     |                   |                              |                                |                              |
>     | ETD            | 2.3 x 1e-1 (2.8 x 1e-1)          | 2.2 x 1e-1 (1.5 x 1e-1)             | **2.1 x 1e-1 (8.6 x 1e-2)**      |
>     |                   | 1.1 x 1e-1 (3.7 x 1e-1)          | 1.2 x 1e-1 (3.6 x 1e-1)             | **7.3 x 1e-2 (6.3 x 1e-2)**      |
>
>     Across all datasets, COLoKe consistently achieves lower generalization and online error than RC, with the sole exception of VdP oscillator, where RC (best online error) slightly outperforms in online error.
>
>     To provide a point of reference on model capacity, we also computed the number of parameters in COLoKe and derived the number of internal nodes that RC would need to match either the total number of parameters (nodes) or just the number of trainable ones (nodes / trainable), with connection density fixed at 0.1:
>
>     | Dataset       | COLoKe (parameters)        | RC (nodes)         | RC (nodes / trainable) |
>     |---------------|----------------|----------------|----------------|
>     | Single attractor   |  778  | 77  | 388 |
>     | Duffing oscillator |  778  | 77  | 388 |
>     | VdP oscillator     |  778  | 77  | 388 |
>     | Lorenz system      | 835  | 75 |  278 |
>     | ETD                | 3188  | 146 | 531 |
>
>     In summary, our grid search already includes reservoir sizes up to an order of magnitude larger than those required to match COLoKe's parameter count. These preliminary results indicate that COLoKe consistently achieves higher accuracy while maintaining a significantly more compact model architecture.
>
>     As we are not specialists in reservoir computing, we would gladly welcome the opportunity to explore other RC methods, should the reviewer have specific methods or references in mind.
>
> References:
> - [Ref. 1] Trouvain, N., Pedrelli, L., Dinh, T.T. and Hinaut, X. ReservoirPy: an Efficient and User-Friendly Library to Design Echo State Networks. ICANN 2020.
> - [Ref. 2] Sussillo, D. and Abbott L.F. Generating Coherent Patterns of Activity from Chaotic Neural Networks. Neuron 2009.

---

> > ### Comment · Reviewer_4zQ8 · 2025-08-04
> > **review of rebuttal**
> >
> > I am happy with the additional experiments and comparison to reservoir computing benchmarks. I keep my current ratings.

---

> > > ### Author Response · Authors · 2025-08-08
> > >
> > > We thank the reviewer for suggesting the additional analysis, which proved valuable in strengthening our study and enhancing the quality of our work. We also appreciate the time invested in reviewing our work.

---

### Official Review · Reviewer_eTLo · 2025-07-03

**Clarity:** 3
**Significance:** 2
**Originality:** 3
**Rating:** 4
**Confidence:** 2

**Summary:**

This paper proposes an extension of online learning for Koopman embeddings to a conformal online learning framework. The approach is validated on synthetic tasks as well as against several online learning baselines.

**Questions:**

Can the authors clarify the intuition behind the construction of the prediction score set?

The paper claims that the prediction set $C_t$ can be used to enhance the interpretability of the model. Have the authors considered any supporting empirical evidence showing how $C_t$ can enhance the interpretability of the model?

**Ethical Concerns:**

["NO or VERY MINOR ethics concerns only"]

**Final Justification:**

The paper is technically strong and good quality. I find the contributions over existing methods on online time-series predictions compelling and novel. This addressed my concerns with the novelty of the proposed method, which the authors addressed at length—borderline accept, low confidence given my limited exposure to literature on online conformal prediction.

**Limitations:**

The performance gains demonstrated in this paper appear to be largely limited to single-attractor systems, raising questions about the method’s generalizability. Furthermore, the novelty of the proposed prediction score set is difficult to assess in the absence of a detailed comparison to existing conformal or online learning methods. A clearer contextualization of the contribution within the broader literature would significantly strengthen the work.

**Quality:**

3

**Strengths And Weaknesses:**

Quality. The overall technical quality of the paper is adequate. The experiments are sound, though relatively limited in scope.

Clarity. The paper is generally clear and well organized. However, the novelty and significance of the prediction score set could benefit from a more detailed explanation.

Significance. The method shows meaningful improvements on single-attractor systems, but results on broader or more realistic datasets are limited. The improvements in real-world applications are marginal, and the paper does not fully explore where or why the approach might generalize well.

Originality. The use of conformal prediction in an online Koopman learning context appears novel. However, it is difficult to assess the distinctiveness of the proposed prediction score set relative to prior work in online learning and conformal prediction. The paper would benefit from a clearer articulation of what makes this construction fundamentally new, and why it is needed beyond existing techniques. This could be supported by an expanded related work discussion that positions the contribution more precisely within the literature on conformal prediction and online learning.

---

> ### Author Rebuttal · Authors · 2025-07-30
>
> We would like to thank the reviewer for their fruitful comments. We have carefully addressed each concern below and believe our clarifications reinforce both the novelty and practical relevance of our contribution. We hope the reviewer finds these clarifications helpful in reassessing the significance of our work. We would be happy to answer any concerns left.
>
> - [L1] *"The method shows meaningful improvements on single-attractor systems, but results on broader or more realistic datasets are limited. [...]"*
>
>     We thank the reviewer for this comment and would like to clarify that our evaluation covers both single-attractor and more complex multi-attractor (Duffing oscillator) / chaotic (Lorenz system) systems. Our method achieves significant improvements in online error (which is the primary performance metric in online learning) showing **gains of at least one order of magnitude across all five datasets**. This includes both synthetic and real-world settings present in the manuscript.
>
>     To further support the generality of our approach, we conducted additional experiments on two new challenging real-world datasets: (i) EEG recordings with 64 channels from the PhysioNet EEG Motor Movement/Imagery dataset [Dataset 1], and (ii) turbulence measurements from the CASES-99 atmospheric field experiment [Dataset 2]. The online errors, summarized in the table below (with standard deviations in parentheses), confirm that our method remains effective in broader high-dimensional scenarios. These findings will be included in the revised version of the manuscript.
>
>     | Dataset       | COLoKe         | ODMD           | OnlineAE       | OLoKe          | OEDMD          |
>     |---------------|----------------|----------------|----------------|----------------|----------------|
>     | EEG (x1e-3)  | **7.9 (8.5)** | 8.3 (11.3) | 12.4 (18.1) | 9.3 (10.4) | 8.0 (11.1) |
>     | Turbulence (x1e-4)   | **4.7 (5.9)** | 5.5 (7.5) | 6.4 (8.6) | 5.5 (6.1) | 7.3 (8.9) |
>
>
> - [L2] *"The use of conformal prediction in an online Koopman learning context appears novel. [...] positions the contribution more precisely within the literature on conformal prediction and online learning."*
>
>     We thank the reviewer for this important point. Our central contribution is to **leverage conformal prediction as a statistically principled event-trigger mechanism** for online learning, a combination that, to our knowledge, has not been previously explored. Our method is the **first that uses conformal scores to actively drive online model updates**. In what follows, to clarify the novelty of our contribution, we first contrast our approach with prior work on online conformal prediction, then discuss related advances in online non-convex learning, and finally position our contribution with respect to recent online time-series forecasting methods.
>
>     - Prior works in **online conformal prediction**, such as Adaptive Conformal Inference (ACI) [Ref. 1], Conformal PID Control [Ref. 2] and their variants (see [Ref. 3] and references therein), focus on dynamically calibrating prediction sets to maintain coverage under distribution shift as new data arrive—either assuming a fixed underlying model, or, when the model is also updated, doing so without using conformal prediction to guide the updates. Unlike these methods, which recalibrate prediction sets in an online manner, our approach uses the conformity score as an indicator of model reliability and triggers model updates only when it exceeds a dynamically calibrated threshold.
>
>     - In parallel, a significant milestone in **online non-convex learning** is the work by Suggala \& Netrapalli [Ref. 4] who achieves optimal regret even under non-convex, adversarial losses, assuming access to an offline oracle. Later work [Ref. 5] on dynamic regret explores variants to handle changing environment. While they address how to update model parameters every round to minimize regret, they do not address when updates should occur and assume updates happen at every step with a randomized prior.
>
>     - Finally, while recent **online time series forecasting methods** have proposed sophisticated adaptation mechanisms [Ref. 6, Ref. 7], they rely on either ad hoc memory-based heuristics or learned change detection modules, and typically involve black-box neural networks. In contrast, our work combines operator-theoretic modeling with conformal calibration to deliver formal guarantees about prediction quality while limiting unnecessary retraining, thus offering a distinct contribution in both methodology and theory.
>
>     In the paper, we chose to focus our discussions on online Koopman operator learning for clarity. Nevertheless, if the reviewer finds it useful, we would welcome the opportunity to include a paragraph in the introduction positioning our work relative to broader literature.
>
> - [Q1] *"[...] clarify the intuition behind the construction of the prediction score set?"*
>
>     Traditionally, conformal prediction aims to construct a prediction set $C_t$ for the next state $x_t$, given a fixed model. In contrast, our perspective is reversed: given $x_t$, we ask which models produce predictions that fall within $C_t$. In this sense, we repurpose the conformal prediction set as a tool to evaluate the model itself, rather than to quantify uncertainty around its output. Importantly, we do not need to explicitly construct $C_t$ as in standard conformal prediction; it is sufficient to evaluate whether the prediction score falls below a threshold. Accordingly, we define the prediction score set as the set of all acceptable scores attainable by Koopman linear models.
>
> - [Q2] *"The paper claims that the prediction set can be used to enhance the interpretability of the model. [...]"*
>
>     We respectfully believe there has been a misunderstanding, as **we do not claim at any point that the prediction set enhances model interpretability**. The only mention of interpretability in the paper refers to the structure of the lifted Koopman embedding, where part of the original state is preserved. Our use of conformal prediction is strictly focused on quantifying uncertainty in the online learning setting, and we make no interpretability-related claims in the manuscript.
>
>
> References:
>
> - [Ref. 1] Gibbs, I. and Candes, E. Adaptive conformal inference under distribution shift. NeurIPS 2021.
> - [Ref. 2] Angelopoulos, A.N., Candes, E. and Tibshirani, R.J., Conformal PID Control for Time Series Prediction. NeurIPS 2023.
> - [Ref. 3] Ramalingam, R., The Relationship Between No-Regret Learning and Online Conformal Prediction. ICML 2025.
> - [Ref. 4] Suggala A.S. and Netrapalli P., Online Non-Convex Learning: Following the Perturbed Leader is Optimal, ALT 2020.
> - [Ref. 5] Xu Z. and Lijun Zhang L., Online Non-convex Learning in Dynamic Environments, NeurIPS 2024.
> - [Ref. 6] Pham, Q. et al., Learning Fast and Slow for Online Time Series Forecasting, ICLR 2023.
> - [Ref. 7] Lau, YA. et al, Fast and Slow Streams for Online Time Series Forecasting Without Information Leakage, ICLR 2025.
> - [Dataset 1] https://physionet.org/content/eegmmidb/1.0.0/
> - [Dataset 2] https://www.eol.ucar.edu/field_projects/cases-99

---

> > ### Comment · Reviewer_eTLo · 2025-08-04
> > **Official Comment**
> >
> > I would like to thank the authors for providing further clarification. The authors have fully addressed my concerns and clarified my misunderstandings. I intend to raise my score accordingly.

---

> > > ### Author Response · Authors · 2025-08-08
> > >
> > > We thank the reviewer for their reconsideration and for their time invested in reviewing our work. We are grateful that our clarifications helped address your concerns.

---

### Note · Authors · 2025-08-12

Dear Reviewers and Area Chair,

We are pleased that our clarifications during the rebuttal period have either maintained or increased your positive assessments of our work. With the additional numerical experiments (two real datasets, reservoir computing) and analysis (computational speed, frequency of updates), the latest comments indicate that earlier concerns have been fully addressed. These additions will be incorporated into the revised manuscript.

We note that no remaining weaknesses have been identified during the discussion period.

We are confident that our contribution, theoretically rooted in a novel repurposing of conformal prediction, will make a meaningful impact.

Thank you for your time and dedication in reviewing our paper.

Sincerely,

The authors

---

### Decision · Program_Chairs · 2025-09-17

**Decision:**

Accept (poster)

**Comment:**

This paper presents an approach to learning deep Koopman representations of non-linear systems in an online setting. This by itself is not a new goal. However, the authors indicate that it may be desirable to limit the number of model updates (or reoptimization steps) due to reasons concerning computational efficiency and overfitting. To accomplish, the work uses conformal prediction to measure how out of character the state evolution seems to be (based on multi-step prediction) for the current estimate of the model, and thus arrives an adaptive schedule of updating the model.

Although by no means a theory paper, the work also nicely links the conformal coverage guarantee to dynamic regret. In the rebuttal period, the authors present more experimental results that convinced engaged reviewers to increment their score. In my reading, the single remaining negative review rests on two points: the experiments and benchmarking is limited, and the idea of using online/adaptive learning to trigger updates is not new. For the first, the reviewer has not responded to the authors' additional experimental results presented during the rebuttal. We agree with the second point, however, there's value to grounding these model updates in a principled framework like conformal prediction.

Overall, given the positive evaluation of this work from all engaged reviewers, we are happy to recommend this paper for acceptance.